# Bringing Code ALIVE: Optimizing Interactive Frontend Mini-Games via Automated Play and Reinforcement Learning at Scale

Jiajun Zhang [* 1 2]  Yuheng Jing [* 2]  Zeyu Cui [3]  Hao Zheng [3]  Wentao Chen [3]  Kaixin Li [3]  Jiaxi Yang [4]
Tianbao Xie [3]  Zeyao Ma [3]  Tianyi Bai [3]  KaShun Shum [3]  Lei Zhang [4]  Kai Li [2]  Jian Cheng [2]  Zilei Wang [1]
Qiang Liu [2]  Liang Wang [2]  Junyang Lin [3]  Binyuan Hui [3]

## Abstract

The rapid evolution of Large Language Models (LLMs) has empowered even non-programmers to create visually appealing frontend mini-games with a single instruction. However, open-source models significantly lag behind proprietary counterparts in this domain. The core bottleneck is the lack of an evaluation mechanism that balances reliability with scalability, as existing methods either fail to verify dynamic interactivity or incur prohibitive computational costs. To bridge this gap, we introduce **ALIVE** (**A**ligning **LL**Ms via **I**nteractive **V**isual **E**xecution), a high-throughput framework that leverages one-shot planning and DOM-based analysis to automatically evaluate generated games at scale. Extensive experiments demonstrate that ALIVE significantly outperforms static judge baselines in identifying functional flaws while remaining orders of magnitude more efficient than GUI agents. Functioning as a scalable 'pre-flight' evaluation layer, it curates high-quality data for Supervised Fine-Tuning (SFT) and provides a consistent reward signal for Reinforcement Learning (RL). We leverage this pipeline to train **ALIVE-Coder**, a model achieving superior performance in interactive frontend generation. To the best of our knowledge, our work offers the first scalable path to evaluate and optimize interactive code, substantially advancing open-source capabilities.

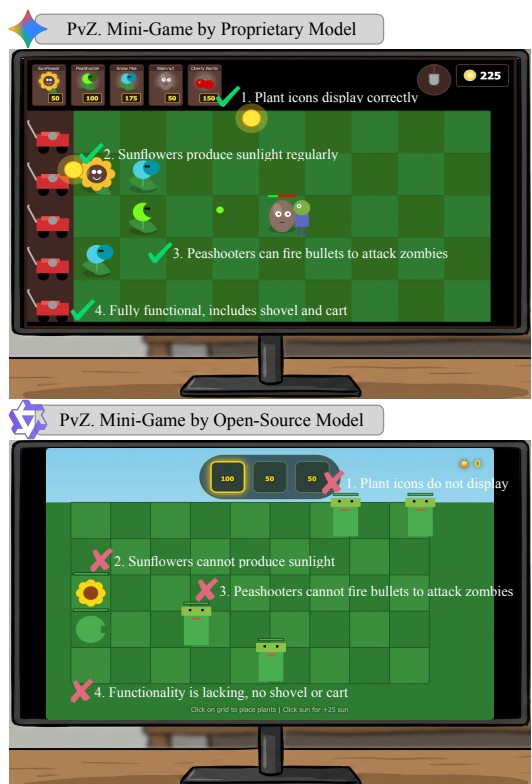

*Figure 1.* **Comparison of generated "Plants vs. Zombies" mini-games.** The proprietary model (Top) produces a fully functional game with correct rendering, resource logic, and interaction mechanics. In contrast, the open-source model (Bottom) fails to render essential assets (e.g., icons) and lacks core interactivity (e.g., shooting), illustrating the significant capability gap.

---

[*]Equal contribution  [1]University of Science and Technology of China, Hefei, China [2]Institute of Automation, Chinese Academy of Sciences, Beijing, China [3]Alibaba Group, Hangzhou, China [4]Shenzhen Institute of Advanced Technology, Chinese Academy of Sciences, Shenzhen, China. Correspondence to: Zeyu Cui <zeyu.czy@alibaba-inc.com>, Liang Wang <wangliang@nlpr.ia.ac.cn>.

*Proceedings of the 43rd International Conference on Machine Learning*, Seoul, South Korea. PMLR 306, 2026. Copyright 2026 by the author(s).

## 1. Introduction

The rapid evolution of Large Language Models (LLMs) has transformed code generation from developer assistance to "vibe coding," where non-programmers create sophisticated web applications via natural language. Leading proprietary models, such as Gemini (Team, 2024) and Claude (Anthropic, 2023), excel in this domain, consistently producing playable, visually appealing applications. In contrast, open-source models like Qwen (Hui et al., 2024a) and GLM (Team et al., 2025a) struggle. Although capable on standard

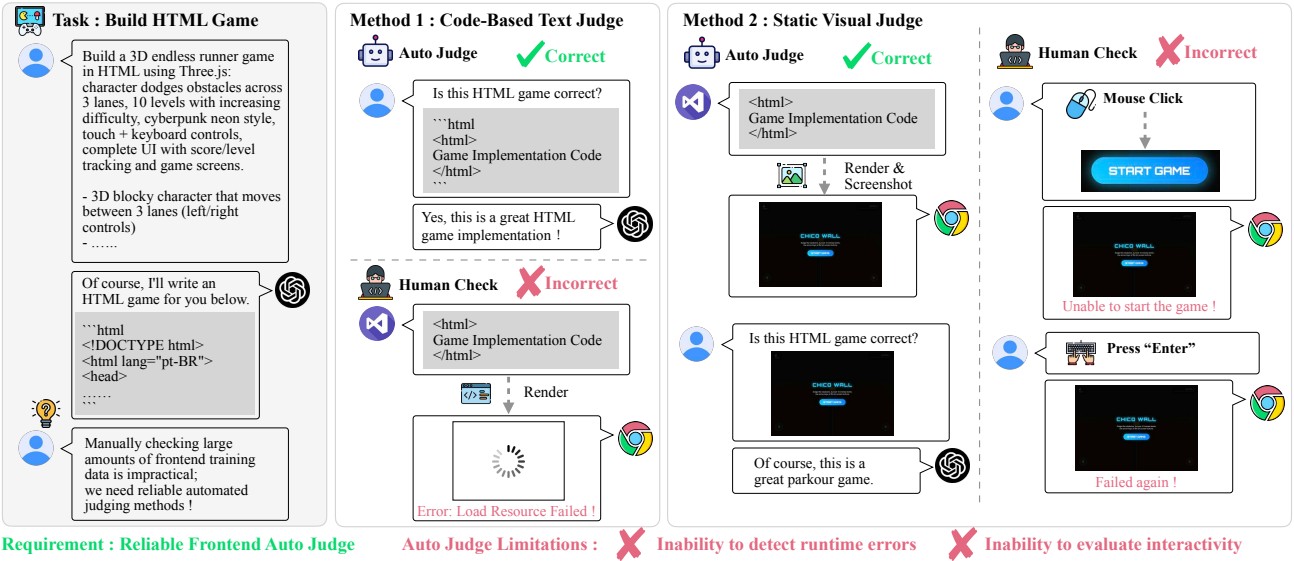

*Figure 2.* **Limitations of static evaluation paradigms.** Code-based judges (Middle) often approve syntactically correct code that fails at runtime (e.g., resource loading errors). Similarly, static visual judges (Right) relying on single-frame screenshots cannot verify interactivity, failing to detect unresponsive controls (e.g., game fails to start). Both approaches frequently yield false positives for non-functional applications.

benchmarks like SWE-Bench (Jimenez et al., 2023b) and LiveCodeBench (Jain et al., 2024), they lag significantly behind in generating interactive frontend applications. Fig. 1 highlights this gap: while Gemini produces a fully playable "Plants vs. Zombies" game, Qwen is functionally broken.

The primary bottleneck impeding open-source advancement lies in the evaluation mechanism. *Current methods fail to balance reliability with scalability.* ❶ **Automated judges lack the intrinsic reliability for interactive tasks.** Static paradigms (e.g., ArtifactsBench (Zhang et al., 2025b)) relying on code analysis or single-frame screenshots frequently miss runtime errors and unresponsive controls (Fig. 2). Furthermore, simple dynamic heuristics (e.g., Monkey Testing) are insufficient; while they trigger basic runtime states, they lack the semantic understanding to verify specific functional requirements (e.g., mechanics logic), often yielding false positives for broken applications (Liu et al., 2025). ❷ **Accurate evaluation methods incur excessive overhead.** Human annotation is prohibitively expensive, and Graphical User Interface (GUI) agents (Zhang et al., 2025a), while capable, suffer from high latency and token costs due to multi-turn interactions. This renders them unsuitable for synthesizing massive training datasets or calculating online Reinforcement Learning (RL) rewards.

To bridge this gap, we propose **ALIVE** (**A**ligning **L**LMs via **I**nteractive **V**isual **E**xecution). ALIVE is a high-throughput evaluation framework designed for scale. Unlike costly iterative agents, ALIVE introduces a verification-oriented one-shot planning mechanism. It analyzes the Accessibility Tree and Document Object Model (DOM) to synthesize comprehensive interaction trajectories in a single inference

pass. By executing these actions and analyzing state transitions, ALIVE assigns a quality score grounded in both functional verification and visual aesthetics. Experiments show that ALIVE achieves higher correlation with human evaluators than static judges, while maintaining a computational cost orders of magnitude lower than GUI agents.

We validate ALIVE through a comprehensive preliminary study, showing it achieves evaluation parity with GUI agents while reducing token consumption and latency by over $10\times$. Leveraging this scalable signal, we curate a large-scale dataset for Supervised Fine-Tuning (SFT) and employ ALIVE as a reward function for Group Relative Policy Optimization (GRPO) (Guo et al., 2025). Notably, this execution-based feedback effectively mitigates reward hacking common in static judges. The resulting model, ALIVE-Coder, demonstrates superior performance in generating interactive frontend mini-games. While ALIVE does not replace exhaustive human testing for pixel-perfect dynamics, it serves as a critical "pre-flight" verification layer. It represents the only currently scalable framework combining visual perception with functional verification, offering a necessary middle ground to enhance frontend capabilities until high-fidelity GUI agents become computationally viable. Our contributions are summarized as follows:

- We propose ALIVE, the first evaluation framework that decouples reasoning from execution via one-shot planning. It establishes a cost–reliability trade-off between evaluation cost and reliability, bridging the gap between static analysis and costly GUI agents.

- We leverage ALIVE as a scalable "pre-flight" verification layer to synthesize a large-scale, high-quality

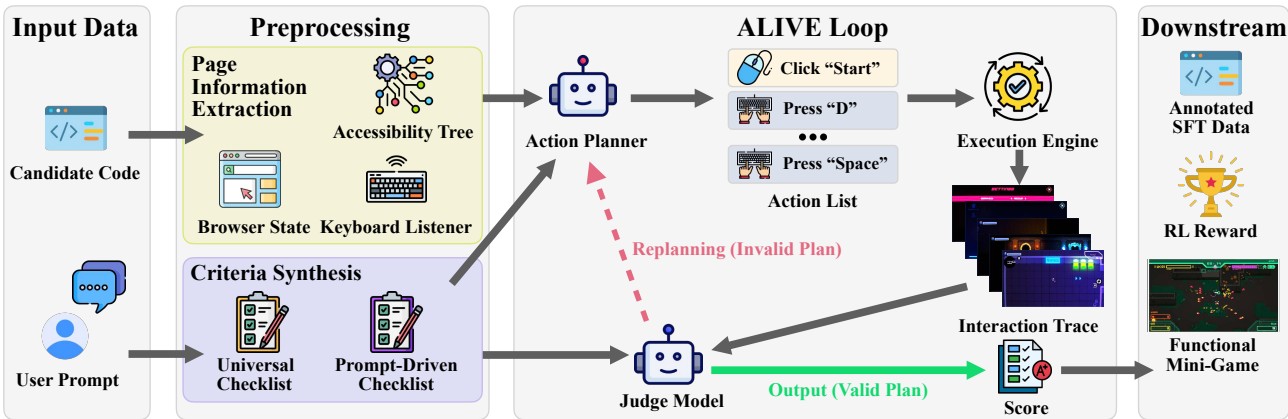

*Figure 3.* **Overview of the ALIVE framework.** The pipeline operates on a "Plan-Execute-Evaluate" cycle. First, the system synthesizes evaluation checklists and analyzes the page structure (DOM/Accessibility Tree) to generate a targeted **Action List**. Next, the **Execution Engine** performs these actions to capture a dynamic **Interaction Trace**. Finally, the **Judge Model** evaluates the trace against the checklists to assign a quality score, utilizing a replanning mechanism to automatically correct invalid interaction plans.

training dataset for frontend games, effectively filtering out functional failures that static judges miss.

- We demonstrate that ALIVE-driven data curation and RL significantly boost interactive abilities of open-source models, verifying that scalable quality feedback is a key driver for advancing frontend generation.

## 2. Methodology

In this section, we present the ALIVE framework. Standard static evaluation metrics fail to capture the dynamic interactivity essential for frontend mini-games, while human evaluation is difficult to scale. ALIVE addresses this by establishing a scalable, automated evaluation pipeline that "plays" the generated applications to assess their quality. The overall architecture of our method is illustrated in Fig. 3. The framework operates on a "Plan-Execute-Evaluate" cycle. Unlike GUI agents that require costly iterative decision-making based on visual feedback for every step, ALIVE leverages the underlying code structure to plan the entire interaction sequence in a single pass. This "one-shot" planning strategy significantly reduces overhead and allows for efficient large-scale evaluation. The pipeline consists of two primary stages: Verification-Oriented Action Planning and Dynamic Execution and Evaluation.

### 2.1. Verification-Oriented Action Planning

The first stage analyzes the frontend mini-game to construct a targeted verification plan. To ensure comprehensive assessment, we first perform a preprocessing step to establish evaluation criteria. For each generated application, we synthesize a task-specific checklist based on the user prompt and combine it with a fixed, universal checklist. This hybrid set of criteria allows us to evaluate both functional correctness (e.g., game mechanics) and aesthetic quality (e.g., visual layout), detailed in Appendix §B.

Next, we utilize a multimodal model to process the rendered page. By analyzing the raw frontend code, Accessibility Tree and DOM, the model understands the interactive elements and structure of the page. Notably, for Canvas-based interactions where explicit DOM nodes are unavailable, we instruct the model to infer approximate click coordinates by analyzing the rendering logic within the raw code. Based on the established criteria and this structural analysis, the model produces the Action List: a structured sequence of interactions (e.g., clicks, key presses) explicitly designed to trigger and verify the items in the checklist. We eschew fixed plans because they enforce rigid constraints (e.g., specific key bindings) that introduce severe bias into training data; instead, our dynamic planning preserves the model's exploration space. This process ensures that the subsequent execution is purposeful and directly grounded in the structure of generated game code (details are in Appendix §C).

### 2.2. Dynamic Execution and Evaluation

In the second stage, we validate the generated code by executing the planned interactions. We utilize a headless browser engine (`Playwright`) to perform the operations defined in the Action List. During this process, we capture the full execution context, including sequential screenshots taken before and after each action, along with execution logs. The Judge Model then analyzes this trajectory against the pre-defined checklists to assign a quality score. Specifically, we employ a rigorous binary evaluation protocol. Each checklist item serves as a binary indicator (0 or 1), verifying the fulfillment of a specific functional requirement based on the interaction trace. With typically 8 to 20 items per task, we calculate the final **ALIVE-Score** by normalizing the sum of satisfied points against the total checklist size.

Crucially, we incorporate a logical feedback mechanism to ensure robustness. After execution, the Judge Model

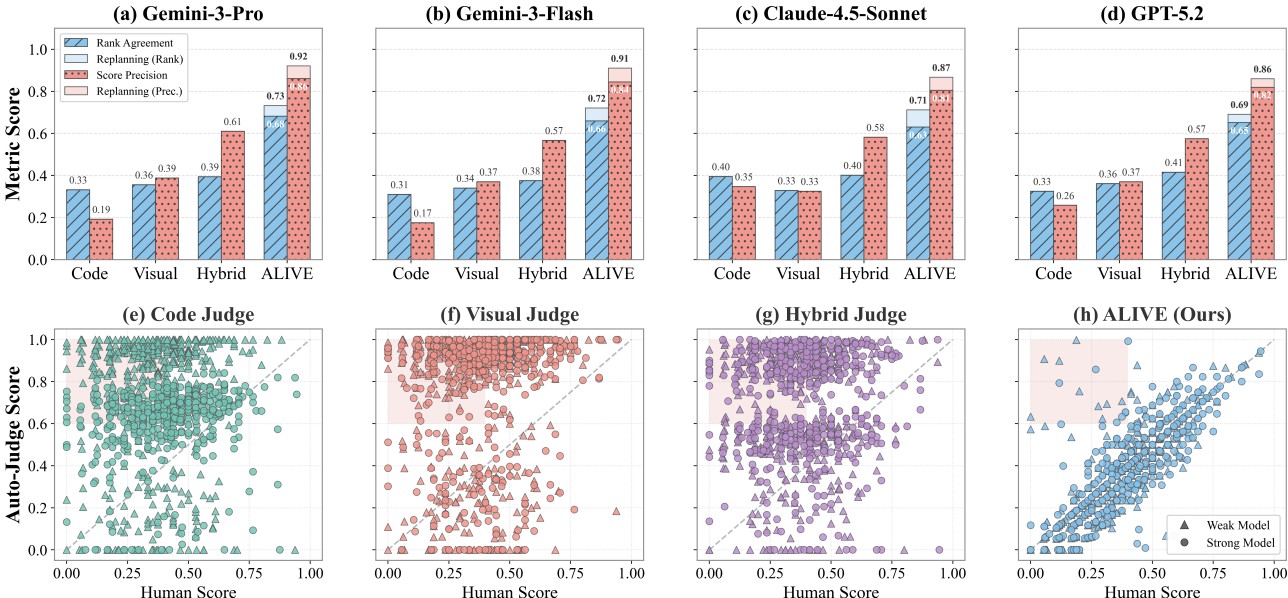

*Figure 4.* **Comparison of Evaluation Methods.** The top row (a-d) displays the Rank Agreement and Score Precision of four evaluation strategies (**Code Judge**, **Visual Judge**, **Hybrid Judge**, and **ALIVE**) across different judge models. ALIVE consistently achieves superior alignment with human judgment. The bottom row (e-h) visualizes the correlation between automated scores and human scores using the averaging outputs of four strongest judge model. While baselines exhibit significant noise and false positives (shaded red regions), ALIVE (h) demonstrates a tight correlation along the diagonal, indicating high reliability.

first assesses whether the generated Action List was reasonable for the current page state. If the actions are deemed infeasible (e.g., attempting to click a non-existent button), the system triggers a regeneration of the Action List to attempt verification again. Only when the action plan is valid does the model proceed to score the mini-game. Detailed implementation settings are provided in Appendix §J.

## 3. Preliminary Study

To validate the necessity of our proposed framework, we conducted a controlled study comparing ALIVE against existing evaluation paradigms. We constructed a dataset of 500 frontend mini-game queries, with solutions generated by `Gemini-3-Pro` (Team, 2024) and `GLM-4.7` (Team et al., 2025b). Ground truth quality scores ($S_{human}$) and pairwise rankings ($R_{human}$) were established by human evaluators. Detailed quality assessment of these human annotations is provided in Appendix §E. We compared four strategies: **Code Judge** (static analysis), **Visual Judge** (screenshot inspection), **Hybrid Judge** (combined), and our **ALIVE** framework. Implementation details are in Appendix §J.

### 3.1. Alignment with Human Judgment

We evaluated the alignment of each method with human preference using four distinct LLMs as backbone judges: `Gemini-3-Pro`, `Gemini-3-Flash`, `Claude-4.5-Sonnet` (Anthropic, 2023), and `GPT-5.2` (OpenAI, 2025). Performance was mea-

sured via: (1) **Rank Agreement:** The consistency in correctly ranking the strong model (`Gemini-3-Pro`) versus the weak model (`GLM-4.7`). (2) **Score Precision:** The proportion of instances where the automated score deviates from the human score by $\leq 0.2$.

**Judge-Agnostic Superiority.** As illustrated in the top row of Fig. 4 (a-d), **ALIVE consistently outperforms all baselines across every judge model**, demonstrating remarkable robustness. Static methods (Code and Visual Judges) exhibit high variance and poor alignment; notably, regardless of the backbone model used, these baselines consistently fail to achieve a Rank Agreement surpassing 0.5. In contrast, ALIVE maintains a high correlation (Rank Agreement $> 0.70$, Score Precision $> 0.85$) across the board. This robust performance confirms that relying solely on static analysis or single-frame visual inspection is insufficient. Instead, our results demonstrate that **integrating dynamic interactivity with visual assessment** is the key to achieving alignment with human perception.

### 3.2. Score Distribution Analysis

To assess the stability of our metrics, we further analyzed the distribution of automated scores versus human scores by **averaging the outputs across all four backbone models**. The bottom row of Fig. 4 (e-h) presents the scatter plots of these averaged scores. Despite the smoothing effect of model ensembling, the baseline methods (e-g) still exhibit substantial noise and a persistent high rate of false positives (highlighted in red shaded regions), where mod-

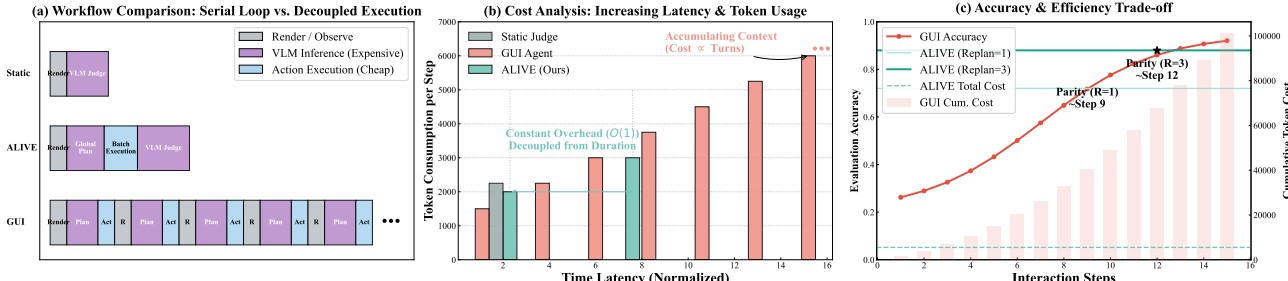

*Figure 5.* **Efficiency, Workflow, and Accuracy Analysis. (a) Workflow:** Unlike GUI Agents which rely on a costly, serial "Observe-Think-Act" loop (bottom), ALIVE decouples reasoning from execution (middle) via one-shot planning. **(b) Cost:** GUI Agents suffer from escalating token consumption (red) as interaction history grows. In contrast, ALIVE maintains near-constant VLM overhead (green) regardless of interaction depth. **(c) Accuracy vs. Steps:** We compared the evaluation accuracy of ALIVE against a GUI Agent running for fixed steps. ALIVE (Replan=3) matches the accuracy of a GUI Agent executing 11-12 serial steps, while ALIVE (Replan=1) corresponds to 9-10 steps. This achieves a **15×** reduction in token cost and **8×** in latency while maintaining parity in verification capability.

*Table 1.* **Cost Analysis for RL Scale.** Calculated for 4,500 prompts × 16 samples using Gemini-3-Flash pricing.

| Method | Samples | Tokens | Time (h) | Cost ($) |
|---|---|---|---|---|
| GUI Agent | 72,000 | 4.8B | 6,048 | 4,800 |
| ALIVE (Ours) | 72,000 | 396M | 264 | 396 |

els assign high scores to games that humans rated poorly. This confirms that static judges systematically hallucinate functionality. E.g., Code Judges tend to bias towards longer, more complex code snippets regardless of actual execution logic, while Visual Judges often overlook functional failures in favor of aesthetic correctness. In contrast, ALIVE (h) shows a strong linear correlation along the diagonal with minimal outliers even after averaging. This validates that ALIVE provides a consistent and robust reward for RL training, unaffected by the variance of specific judge models.

### 3.3. Necessity of Semantic Planning

We benchmarked ALIVE against heuristic execution policies (e.g., Random Monkey) on fully functional games (Appendix §I). Results reveal that heuristics suffer from severe **evidence starvation**: they fail to sustain valid gameplay to trigger core mechanics (e.g., causing instant "Game Over"), resulting in low *Mechanism Exposure Rates* and incorrect penalization of valid games. In contrast, ALIVE's semantic planning ensures interactions are causally linked to verification goals, securing the high-quality visual evidence necessary for reliable assessment.

### 3.4. Efficiency Analysis

Beyond accuracy, computational overhead is a critical constraint for scaling evaluation. We compare the inference cost and performance parity of ALIVE against Static Judges and existing GUI Agent (Zhang et al., 2025a).

**The Cost-Accuracy Trade-off.** GUI Agents rely on a serial "Observe-Think-Act" loop, requiring a fresh rendering and VLM inference for every interaction step (Fig. 5(a)).

While this allows for dynamic correction, it incurs a linear cost penalty ($O(N)$) where token consumption and latency accumulate super-linearly with interaction depth (Fig. 5(b)).

**Performance Parity.** As shown in Fig. 5(c), we benchmarked the evaluation accuracy of ALIVE against a GUI Agent constrained to varying interaction steps. The results demonstrate that ALIVE (with a max of 3 replannings) achieves an accuracy equivalent to a GUI Agent performing 11-12 serial interaction steps. Even with a single replanning attempt, ALIVE matches the performance of 9-10 agent steps. By synthesizing long-horizon action chains in a single pass, ALIVE delivers this high-fidelity verification with approximately **12×** lower token consumption and **22×** lower latency than its agent-based counterpart. As highlighted in Table 1, for large-scale RL training (sampling 16 candidates), ALIVE reduces the serial execution time from over 6,000 hours to 264 hours and slashes the API cost from $4,800 to $396. Detailed performance and cost analyses are provided in Appendix §F.

## 4. Experiments

In this section, we detail our experimental setup and analysis. We first describe the construction of our specialized dataset in Section §4.1. We then present the implementation details of SFT in Section §4.2 and RL in Section §4.3. Finally, we provide a comprehensive analysis of the experimental results in Section §4.4.

### 4.1. Dataset Construction

We constructed the **ALIVE-Mini Game Dataset** to facilitate the training and evaluation of frontend game generation models. To ensure diversity, we collected metadata from multiple public web game platforms. The collected information includes game titles, genre tags, core mechanic keywords, and official screenshots. Leveraging this metadata, we utilized multiple open-source models to synthesize a diverse set of 15,000 frontend mini-game queries. The dataset is partitioned into three distinct subsets: 10,000 samples for

SFT, 4,500 samples for RL, and 500 samples for evaluation, where the evaluation subset is denoted as **ALIVE-Eval**.

For the ALIVE-Eval subset, we established a rigorous ground truth standard. The mini-game authors manually enriched each query with specific implementation details, such as control schemes and layout specifications. Furthermore, based on these details, we manually constructed precise verification checklists to ensuring accurate evaluation. Regarding copyright and ethical considerations, we clarify that game titles are utilized as factual information, and short genre tags constitute generic descriptions that are not subject to copyright protection. All collected data is utilized strictly for academic research purposes and will not be used commercially. To respect intellectual property rights, the raw source data will not be publicly released.

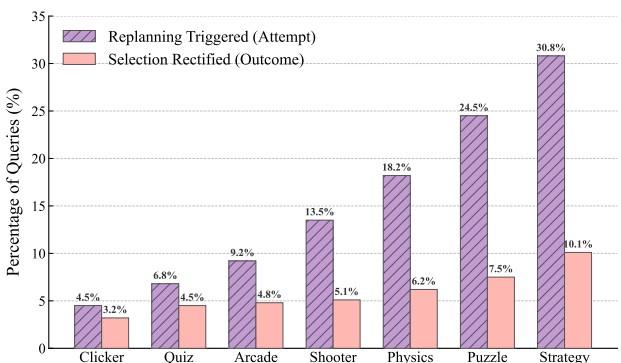

*Figure 6.* **Analysis of Replanning Effectiveness.** The grouped bar chart compares the frequency of replanning attempts (Purple) against successful selection rectifications (Pink) across game genres. While the gap between attempts and rectifications reflects the exploration cost, the rising trend in rectifications for complex genres demonstrates the necessity of dynamic planning for ensuring dataset diversity. (Full results are in Appendix §D)

### 4.2. ALIVE-Driven Supervised Fine-Tuning

**Experimental Setup.** To construct high-quality training data, we utilized the 10,000 queries from the SFT subset. We employed `Gemini-3-Pro` to generate five distinct candidate solutions for each query. We then applied the ALIVE framework to evaluate these candidates. To ensure robust evaluation and minimize false negatives caused by improper planning, we configured ALIVE with a maximum of three replanning attempts for the action list generation. For each query, we selected the response with the highest **ALIVE-Score** to form the final SFT dataset. For the training phase, we selected `Qwen3-Next` as the base model. We performed SFT using Megatron (Shoeybi et al., 2019). The training process consisted of 3 epochs with a global batch size of 128. We utilized a constant learning rate of $7 \times 10^{-6}$ and a cosine decay schedule for weight decay. The resulting model is denoted as **ALIVE-Coder-SFT**.

**Verification of Replanning Strategy.** The impact of this strategy is quantified in Fig. 6. We report two metrics: the frequency of replanning attempts triggered by ALIVE and the proportion of samples where the final selection was actually rectified. As observed, the attempt rate is consistently higher than the rectification rate, indicating a rigorous verification process. Overall, the mechanism successfully rectified 6% of the total dataset. This contribution is particularly pronounced in complex genres (e.g., Puzzle, Strategy), preventing the dataset from becoming biased towards simpler, static pages and preserving structural complexity.

### 4.3. RL with ALIVE-Driven Interactive Feedback

While SFT establishes a strong foundation for syntax and basic logic, it often fails to capture the intricate, long-horizon dynamics required for fully functional gameplay. To bridge this gap, we employ RL to directly optimize the model using our **ALIVE-Score**.

**Experimental Setup.** We initialize our policy with **ALIVE-Coder-SFT** and employ GRPO, a highly efficient PPO (Schulman et al., 2017)-variant designed for reasoning-intensive tasks. The reward is derived exclusively from the ALIVE evaluation pipeline, using `Gemini-3-Flash` as the underlying judge to balance cost and capability. We train on the RL subset (4,500 samples) for above 1 epoch. To ensure training stability and computational efficiency, we utilize a global batch size of 32 with a mini-batch size of 16. We set the learning rate to $2 \times 10^{-6}$ and the context length to 64k to accommodate long interaction traces. To mitigate the training-inference inconsistency inherent in policy gradients over complex code generation, we implement a Truncated Importance Sampling (Yao et al., 2025) mechanism with a ratio clipping upper bound of 5.

**Simplification of Replanning Strategy.** During RL training, we restricted the replanning mechanism to a single iteration . While multi-turn replanning improves evaluation accuracy, it introduces significant computational overhead during reward calculation. Due to limited CPU resources for parallel execution, the marginal accuracy gain from extensive replanning did not justify the extended training duration. This limitation is purely hardware-dependent and can be trivially resolved by scaling the number of CPU workers.

### 4.4. Results and Discussion

#### 4.4.1. EVALUATION DETAILS.

We evaluate our models on two benchmarks: **ALIVE-Eval**, our proposed frontend mini-games evaluation dataset, and **ArtifactsBench** (Zhang et al., 2025b), a standard benchmark focusing on static visual quality (We use its **Game** subset). We compare against state-of-the-art proprietary models (`GPT-5.2`, `Claude-4.5-Sonnet`, `Gemini-3-Pro`) and strong open-source baselines

(`DeepSeek-V3.2` (DeepSeek-AI & etc., 2024), `GLM-4.7`, `Qwen3-Coder-Plus` (Hui et al., 2024a)). See implementation details in Appendix §G.

### 4.4.2. MAIN RESULTS

The comparative results are summarized in Table 2.

*Table 2.* **Main Results on Frontend Generation Benchmarks.** We compare our models against state-of-the-art proprietary (Gray background) and open-source models. Scores on ALIVE-Eval are computed using our automated pipeline, while ArtifactsBench scores are reported from their standard static evaluation.

| Model | ALIVE-Eval | ArtifactsBench-Game |
|---|---|---|
| Gemini-3-Pro | 39.2 | 57.1 |
| Claude-4.5-Sonnet | 37.7 | 56.7 |
| GPT-5.2 | 40.3 | 58.4 |
| DeepSeek-V3.2 | 27.7 | 57.6 |
| GLM-4.7 | 28.1 | 56.2 |
| Qwen3-Coder-Plus | 25.9 | 60.9 |
| Qwen3-Next | 25.4 | 49.2 |
| **ALIVE-Coder-SFT** | 29.1 | 58.4 |
| **ALIVE-Coder-RL** | **40.5** | **61.2** |

**SFT and RL Performance.** Table 2 shows that **ALIVE-Coder-SFT** significantly improves upon the base model `Qwen3-Next` (Yang et al., 2025), raising the ALIVE-Eval score from 25.4 to 29.1 (+3.7) and the ArtifactsBench score from 49.2 to 58.4 (+9.2). This substantial gain confirms that data curation via ALIVE effectively filters low-quality samples, enabling the model to learn robust implementation logic. Results imply that RL further amplifies these gains. **ALIVE-Coder-RL** achieves dominant performance across both metrics. On ALIVE-Eval, it attains a score of **40.5**, surpassing all open-source baselines and outperforming the strongest proprietary model, `GPT-5.2` (40.3). Notably, on the static ArtifactsBench, it also reaches a state-of-the-art score of **61.2**, slightly edging out `Qwen3-Coder-Plus` (60.9). This demonstrates that optimizing for dynamic interactivity via ALIVE not only masters functional correctness but also enhances visual quality.

**Human Evaluation.** To corroborate these findings, we conducted a human evaluation on the **ALIVE-Eval**, following the same protocol described in §3 to assess all model outputs and compute pairwise preferences. As shown in Fig. 7, ALIVE-Coder-RL achieves a dominant win rate (>90%) against the base model and maintains a substantial margin over strong open-source competitors like DeepSeek-V3.2, confirming that the improvements in automated metrics translate directly to perceptible gains in user experience.

### 4.4.3. RL TRAINING DYNAMICS AND ANALYSIS

**Performance and Robustness.** We compare our ALIVE-driven RL against two baselines: RL driven by the **Visual**

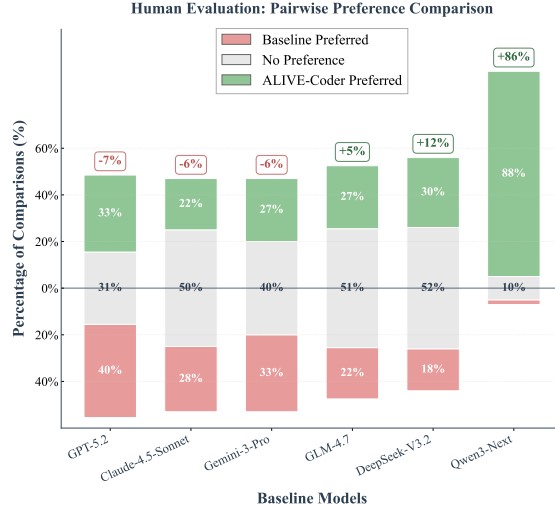

*Figure 7.* **Blinded Pairwise Human Evaluation.** The vertical diverging chart illustrates user preference for ALIVE-Coder-RL. The model demonstrates competitive performance comparable to top-tier proprietary models.

**Judge** and the **Hybrid Judge**. As illustrated in Fig. 8 (Left), **ALIVE-Coder-RL** (blue curve) demonstrates a steady and significant improvement in test set performance, rising from an initial score of $\sim$0.25 to over 0.40. This confirms that dynamic interaction provides a generalizable learning signal that transcends mere memorization.

**Analysis of Reward Hacking.** A critical insight from our experiments is the susceptibility of static evaluators to reward hacking, a phenomenon captured in training:

- **The Verbosity Trap (Hybrid Judge)**: The Hybrid Judge baseline (purple curve) exhibits a pathological failure mode. While its training score (Fig. 8, Middle) remains high, its test score (Fig. 8, Left) collapses. The cause is revealed in Fig. 8 (Right): the response length explodes, doubling from $7k$ to over $14k$ tokens. The model learned to exploit the static code analyzer's bias toward complex, verbose structures, 'spamming' syntactically valid but functionally incoherent code to maximize the reward. This results in a 'high-confidence' failure where the agent optimizes for code length rather than game functionality.

- **The Hallucination Trap (Visual Judge)**: The Visual Judge baseline (red curve) suffers from noisy rewards. By relying on static screenshots, the reward signal fails to penalize frozen games or non-interactive UI elements, leading to stagnation in real-world performance.

In stark contrast, **ALIVE-Coder-RL** mitigates reward hacking by grounding the optimization objective in **causal execution verification**. Unlike static proxies that are susceptible to superficial correlations—such as the 'verbosity bias' of text judges or the aesthetic priors of visual judges—ALIVE's reward is strictly coupled to **observ-**

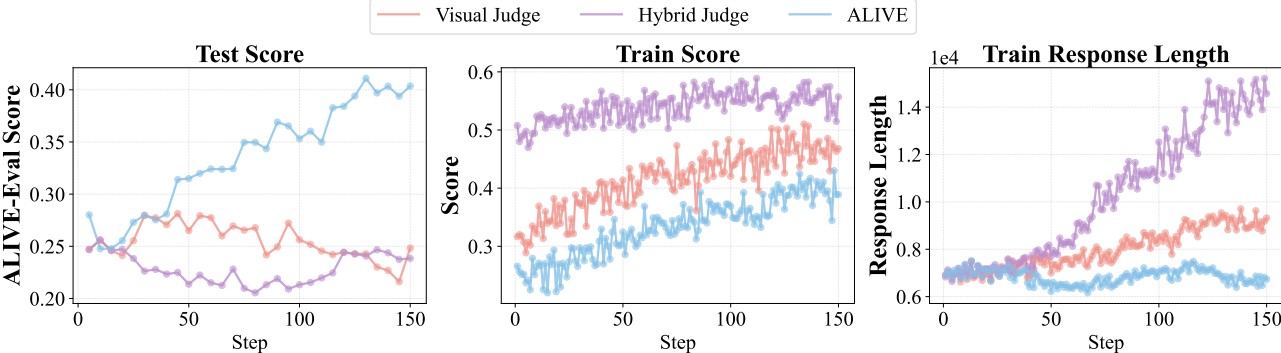

*Figure 8.* **Analysis of RL Training Dynamics and Reward Hacking.** We compare policies trained with three different reward signals: Visual Judge (Red), Hybrid Judge (Purple), and our ALIVE framework (Blue). **(Left)** On the held-out test set, only the ALIVE-driven policy achieves sustained performance gains, whereas baselines stagnate or degenerate. **(Middle & Right)** Training logs reveal the susceptibility of static judges to reward hacking. The Hybrid Judge falls into a '*verbosity trap*,' where the agent learns to maximize rewards by exploding the response length (Right) rather than improving functionality. Similarly, the Visual Judge provides noisy signals that fail to guide meaningful optimization. In contrast, ALIVE provides a robust execution-based signal, ensuring that improvements in training rewards translate effectively to test-time functional correctness without inducing unnecessary verbosity.

**able state changes** driven by interaction (e.g., a keypress must trigger a verified coordinate update). This mechanism imposes a structural penalty on non-functional complexity: since the reward is strictly tied to successful dynamic checks, increasing code length without enhancing interactivity yields no marginal gain and explicitly increases the risk of runtime rendering failures. Empirically, this robust alignment is validated by the **stable response length** (Fig. 8, Right, blue curve), which accompanies sustained improvements in functional correctness. These results underscore a principle: stabilizing RL for frontend generation requires shifting from static proxy metrics to **interactive, execution-based verification**, ensuring the policy optimizes for genuine utility rather than exploiting metric loopholes.

## 5. Related Work

**Code Generation & Evaluation.** Recent advancements in LLMs have revolutionized code generation, evolving from foundational models (Roziere et al., 2023; Lozhkov et al., 2024; Guo et al., 2024; Hui et al., 2024b; Cao et al., 2026) to autonomous agents capable of resolving repository-level issues (Yang et al., 2024; Wang et al., 2024). However, existing benchmarks predominantly rely on static unit tests (Jimenez et al., 2023a) or text-based metrics, which fail to capture the visual and interactive dynamics essential for frontend development. ALIVE bridges this gap by introducing automated visual execution to verify correctness in interactive scenarios.

**Agentic Judges & Generalist Control.** To address static metric limitations, 'Agent-as-a-Judge' paradigms have emerged (Zhuge et al., 2024; Liu et al., 2025; Wang et al., 2026). Most closely related to our work is the Computer-Use Agents as Judges framework (Lin et al., 2025); however, while it often incentivizes 'destylization' to assist agent

perception, ALIVE focuses on verifying 'human-friendly' visual richness. Furthermore, ALIVE employs a one-shot planning mechanism to decouple reasoning from execution, avoiding the high latency of serial 'observe-think-act' loops inherent in generalist agents (Wang et al., 2025b;a). By providing scalable evaluation infrastructure, ALIVE complements these generalist agents, optimizing the *generation* of the interactive environments they navigate.

*A comprehensive related work is provided in Appendix A.*

## 6. Limitations

**Planning Precision in Real-Time Dynamics.** One-shot planning may struggle with highly real-time genres (e.g., parkour) where precise trajectory prediction is difficult. However, by capturing dynamic intermediate states, ALIVE still yields much higher evaluation accuracy than static baselines. See detailed failure rates by genre in Appendix §D.

**Concurrency Bottlenecks.** The execution engine relies on `Playwright`, where the underlying serial pipe architecture limits high-concurrency scaling due to resource conflicts. We address this via Docker isolation, though each container is physically limited to supporting approximately 30 parallel browser instances (Appendix §G.0.1).

## 7. Conclusion and Future Works

The advancement of open-source models in **automated frontend generation** has been stalled by an evaluation mechanism that is either too static to be reliable or too costly to scale. **ALIVE** breaks this deadlock by introducing a high-throughput verification layer that decouples reasoning from execution. By establishing a cost–reliability trade-off, ALIVE transforms the verification of interactive code from a computational bottleneck into a scalable data

engine. Our experiments confirm that grounding evaluation in dynamic execution is decisive for closing the capability gap with proprietary models, even when implemented via efficient one-shot planning. While currently a "pre-flight" filter, ALIVE paves the way for future research. We plan to extend this work by developing sophisticated yet cost-efficient GUI agent frameworks and integrating stronger judge models to address the complex, real-time stochastic dynamics required for high-fidelity interaction.

## Impact Statement

This work advances the field of automated software engineering by bridging the critical gap between static code generation and dynamic interactive evaluation. By establishing a scalable framework for verifying frontend applications, our research contributes to the democratization of software development, empowering non-experts to create functional interactive media. Furthermore, by rigorously penalizing non-functional generation through execution-based feedback, our approach helps mitigate the risks of hallucinated code, promoting the deployment of more reliable and trustworthy AI systems.

## Acknowledgments

This work is supported by National Key Research and Development Program (2023YFC3305203), National Natural Science Foundation of China (92570204, 62576339).

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

# A. Extensive Related Work

**LLMs for Code Generation.**   The landscape of automated software engineering has been fundamentally transformed by the advent of Large Language Models (LLMs), shifting the paradigm from simple syntactic completion to semantic reasoning. Foundational models, trained on trillions of code tokens, have established robust baselines for multilingual code generation. Notable examples include CodeLlama (Roziere et al., 2023), StarCoder (Lozhkov et al., 2024), DeepSeek-Coder (Guo et al., 2024), and Qwen-Coder (Hui et al., 2024b). To enhance the capability of these models in handling context-dependent scenarios, training objectives have evolved beyond causal language modeling to include Fill-in-the-Middle (FIM) strategies (Bavarian et al., 2022; Zhang et al., 2026), which enable models to utilize bidirectional context for precise code infilling. Beyond pre-training, recent research emphasizes instruction tuning to align models with complex human intents and real-world development workflows. A critical advancement in this domain is the synthesis of high-quality, verifiable engineering data. For instance, Zhang et al. (2025c) demonstrated that training on synthesized Test-Driven Development (TDD) data significantly improves a model's ability to produce functionally correct and test-compliant code. These improvements in underlying models have catalyzed the emergence of autonomous coding agents, such as SWE-Agent (Yang et al., 2024) and OpenDevin (Wang et al., 2024). Unlike standalone code generators, these agents employ iterative planning and tool use to resolve repository-level issues, a capability rigorously benchmarked by SWE-Bench (Jimenez et al., 2023a). However, despite these strides in functional correctness and logic verification, existing paradigms exhibit a critical limitation: they primarily rely on static unit tests or text-based similarity metrics (e.g., BLEU, CodeBLEU) for evaluation. This text-centric approach is insufficient for the emerging domain of interactive frontend development—often referred to as 'vibe coding'—where the output is visual and dynamic rather than purely logical. In such scenarios, a code snippet may pass all unit tests yet fail to render a playable or visually coherent interface. To bridge this gap between logical correctness and visual viability, we propose ALIVE, a framework designed to verify and optimize interactive code through automated visual execution and feedback.

**Agentic Frontend Judge and Automated Evaluation.**   The inadequacy of static evaluation metrics (e.g., BLEU or exact code matching) for assessing interactive software has precipitated a paradigm shift toward Agent-as-a-Judge' (Zhuge et al., 2024). In this paradigm, autonomous agents replace passive scripts to execute, observe, and evaluate system performance in dynamic environments. This approach has been rigorously applied in automated web testing to handle modern dynamic applications; for instance, Judge (Liu et al., 2025) addresses the state explosion' problem in GUI crawling by employing a Merge-and-Classify' strategy, which utilizes contrastive learning and SVMs to abstract diverse DOM structures into unified states for efficient exploration. In the domain of generative frontend development, evaluations must also account for the iterative nature of design. FronTalk (Wu et al., 2025) benchmarks conversational code generation by integrating agentic critiques via AceCoder, a retrospective judge that mitigates the forgetting issue' in multi-turn dialogues by verifying current outputs against historical constraints.

Most closely related to our work is the framework proposing Computer-Use Agents (CUAs) as Judges (Lin et al., 2025), which establishes a collaborative loop where agents navigate and critique generated user interfaces to guide refinement. However, a critical divergence in optimization objectives exists: while Lin et al. (2025) often incentivizes destylization'—stripping away complex visual elements to create simplified, agent-friendly' interfaces optimized for machine perception—ALIVE focuses on verifying that the generated mini-games remain human-friendly,' prioritizing rich visual rendering and complex interactivity. Furthermore, regarding computational efficiency, standard GUI agents operate on a serial observe-think-act' loop (Zhang et al., 2024; Nguyen et al., 2024). This paradigm incurs high latency and token costs ($O(N \times L)$) as the context window grows with interaction history. In contrast, ALIVE introduces a 'one-shot planning' mechanism that decouples reasoning from execution. By generating a comprehensive action schedule in a single inference pass ($O(1)$) and offloading execution to a lightweight engine, ALIVE offers a significantly more scalable and cost-effective solution for providing dense reward signals in Reinforcement Learning.

**LLMs for Games and Generalist Computer Control (GCC).**   The intersection of Large Language Models (LLMs) and game simulation serves as a crucible for achieving Generalist Computer Control (GCC), testing an agent's ability to reason over long horizons in partially observable environments. Comprehensive surveys on GUI agents (Zhang et al., 2024; Nguyen et al., 2024) highlight the field's transition from specialized API-based interactions to universal Human Interface Device (HID) actions, such as keyboard and mouse events based on visual inputs. Despite this progress, generalist Vision-Language Models (VLMs) continue to struggle with the temporal dynamics and strict latency requirements of gaming. This limitation is evidenced by VideoGameBench (Zhang et al., 2025a), where state-of-the-art models exhibit severe performance gaps—achieving less than $1\%$ success on full benchmarks—necessitating a 'lite' evaluation mode that

pauses environments to accommodate slow inference.

To bridge this capability gap, recent foundation models have adopted specialized architectures. Game-TARS (Wang et al., 2025b) employs a unified action space and a sparse-thinking' strategy, planning high-level moves intermittently to manage inference costs while mastering diverse game genres. Similarly, UI-TARS-2 (Wang et al., 2025a) leverages multi-turn Reinforcement Learning with a data flywheel'—synthesizing and filtering trajectories—to achieve human-level performance across both GUI and game tasks. ALIVE complements these advancements by addressing the generative side of the GCC ecosystem. Whereas Game-TARS and UI-TARS focus on the agent-as-a-player, optimizing policies to conquer existing games, ALIVE provides the necessary evaluation infrastructure to train models (like ALIVE-Coder) capable of creating high-quality, bug-free, and functionally complex interactive environments that these generalist agents can subsequently navigate and master.

## B. Checklist Synthesis Details for Verification-Oriented Action Planning

### B.1. Universal Checklist (Complete Item Set)

For every mini-game, ALIVE evaluates a fixed *universal checklist* to ensure basic playability and presentation quality independent of the specific user request:

1. **Lifecycle correctness.** Verify the game functions correctly through its full lifecycle: ensure the game loads without errors, transitions smoothly from the start screen to gameplay, correctly triggers win/loss states, and allows for restarting without freezing or crashing.

2. **Layout visibility.** Verify the visual layout is consistent and clear: ensure all UI elements (text, buttons, HUD) and game objects are fully visible within the viewport, properly aligned, and free from unintended overlapping, cropping, or rendering issues.

3. **Logic/physics plausibility.** Verify game logic and physics appear natural and consistent: ensure objects move and interact according to expected rules (e.g., gravity, collision detection) without glitching, clipping through boundaries, or exhibiting erratic, non-intuitive behavior.

4. **Visual cohesion & legibility.** Verify the visual presentation is cohesive and legible: ensure the art style is unified, text is easily readable against the background, and the color palette is harmonious (avoiding jarring, painful contrasts or "developer art" placeholders).

5. **Text correctness.** Verify text and displayed content are error-free: ensure all on-screen text (menus, instructions, scores) is spelled correctly, grammatically sound, and free from placeholder text (e.g., "Lorem Ipsum") or encoding glitches.

### B.2. Prompt-Driven Checklist Generation

**Goal.** In addition to the universal checklist, ALIVE synthesizes a *prompt-driven checklist* that translates the user request into a set of binary, execution-verifiable requirements. Each item is written to be checkable from the interaction trace (screenshots before/after actions, DOM/Accessibility Tree snapshots, and execution logs).

**Generation prompt (system + user template).** We use a single-call instruction to produce a checklist in a constrained format and explicitly discourage non-verifiable or subjective criteria.

> **System:** You are an evaluation designer for interactive HTML mini-games. Convert the user's request into a verification checklist whose items are *binary* and *testable* via browser interaction traces (clicks/keypresses/touches), DOM/accessibility tree inspection, and screenshots. Avoid vague wording (e.g., "nice", "fun", "smooth"). Each item must describe an observable pass/fail condition.
>
> **User:** Given the user prompt below, output **8–20** checklist items.
>
> **User Prompt:** {USER_PROMPT}
>
> **Output format (strict):**
>
> C1: {one sentence}
> C2: {one sentence} . . .
>
> **Constraints:** (1) At least 3 items must concern core mechanics (controls, state updates, collisions, scoring/resources, win/loss).
> (2) At least 2 items must concern UI/HUD/state display correctness (score, timer, level, health, etc.), if applicable.
> (3) If the prompt specifies controls (e.g., WASD, arrows, touch), include an item that verifies each specified control works.
> (4) If the prompt specifies levels/difficulty, include items verifying progression and parameter change across levels.
> (5) Do not mention implementation details (e.g., libraries) unless explicitly required by the prompt and verifiable.

**Few-shot examples (prompt–checklist pairs).** We include a small number of demonstrations where each checklist line is the *title* only (as used by our binary rubric).

**Prompt–Checklist Pair A (Mad Burger 2: Winter Edition (Invincible)).** **Prompt (abridged):** Design a 2D side-scrolling physics-based launcher game where Chef Charlie launches a hamburger via a power gauge, applies mid-air "Sauce" boosts by mouse clicks (limited gauge), bounces and collides with terrain/objects/campers for bonuses, ends the run when the burger stops, awards money by distance and interactions, and provides a shop to upgrade launch power/aerodynamics/bounciness/sauce capacity/magnet. Include cheat keys: 1 refills sauce and 2 adds money. Visual theme: colorful winter campsite with snow and pine trees; character is a chef with ushanka and red scarf; HUD shows distance, sauce gauge, speedometer, and money.

**Example 1:** *Mad Burger 2: Winter Edition (Invincible)*

C1: Verify the game loads with a 'Play' button and transitions to the Launch phase.

C2: Verify clicking the mouse stops the power meter and launches the burger.

C3: Verify the burger is affected by gravity and eventually lands on the ground.

C4: Verify clicking the mouse in mid-air consumes the 'Sauce' gauge and applies a velocity boost to the burger.

C5: Verify pressing the '1' key refills the Sauce gauge (Invincible mechanic).

C6: Verify pressing the '2' key increases the player's Money balance (Invincible mechanic).

C7: Verify the run ends when velocity reaches zero, displaying a results screen with distance traveled.

C8: Verify the Shop allows purchasing upgrades which permanently affect subsequent launches (e.g., higher initial speed).

C9: Verify the player character is a chef wearing a winter hat (ushanka) and red scarf.

C10: Verify the background features a winter theme with snow and pine trees.

C11: Verify the projectile is a stylized hamburger.

C12: Verify a visual gauge is present for the Sauce/Fuel level.

**Prompt–Checklist Pair B (Snake Falls).** **Prompt (abridged):** Create a 2D side-view puzzle/logic game with grid-based movement controlling a pink blocky snake that must collect an orange fruit to unlock an exit portal, then reach the spinning rainbow portal to complete the level. Fail states: falling into bottom water or touching rotating gear hazards triggers restart. Gravity applies with "support/grip" logic: the snake falls only when no segment is supported. UI: level select screen with 24 stages; restart and menu buttons in top corners. Visuals: bright flat vector style with sky, clouds, triangular mountains, floating islands with brown earth and green grass.

**Example 2:** *Snake Falls*

C1: The game must load to a Level Selection screen with at least 24 selectable stages.

C2: Arrow keys must move the snake's head one grid unit at a time, with the body following.

C3: The Exit Portal must remain locked or inactive until the player collects the Orange Fruit.

C4: Touching a grey gear obstacle must immediately trigger a restart/fail state.

C5: The gravity system must check if the snake is supported; if no segments touch the ground, the snake must fall.

C6: Falling off the bottom of the screen into the water must trigger a Game Over.

C7: Entering the Exit Portal after collecting the fruit must load the next level.

C8: The game must feature a side-view 2D perspective with a blue sky and mountain background.

C9: The player character must be depicted as a pink, blocky snake with a face.

C10: Terrain must appear as floating islands with brown earth and green grass tops.

C11: The Exit Portal must be a colorful, spinning rainbow wheel.

C12: UI buttons (Restart, Home) must be positioned in the top corners of the screen.

**Prompt–Checklist Pair C (Telekinetic Robot Arena).** **Prompt (abridged):** Create a 3D third-person physics brawler where the player controls a blue segmented robot in a grid-textured test chamber, using a "gravity gun" telekinesis mechanic to grab distant physics objects (RMB/scroll) to a hold point and throw them (LMB) to damage enemies with ragdoll. Enemies spawn, chase, and deal melee damage; red barrels explode on impact. UI includes a top-left kill counter and bottom-left vertical green segmented health bar; win when kill counter reaches target; lose on death. Visual style: low-poly, clean, bright prototyping aesthetic with white grid floor.

**Example 3: *Telekinetic Robot Arena***

C1: Verify the game loads into a 3D environment with a controllable character.

C2: Verify WASD moves the robot and Mouse rotates the camera.

C3: Verify the 'Gravity/Suck' mechanic: Pressing the input must pull a distant loose object to the player's hold point.

C4: Verify the 'Throw' mechanic: Pressing fire while holding an object must launch it forward with physics force.

C5: Verify collision logic: Thrown objects must collide with enemies and cause damage/death (triggering ragdoll state).

C6: Verify enemy AI: Enemies must spawn and actively chase the player to deal melee damage.

C7: Verify explosive barrels: Throwing a red barrel must result in an explosion event that damages nearby entities.

C8: Verify Win Condition: The level ends or displays a success message when the kill counter reaches the total target.

C9: Verify the art style is low-poly 3D with a bright, clean aesthetic.

C10: Verify the player character is a blue segmented robot.

C11: Verify the environment uses a white grid floor texture.

C12: Verify the HUD includes a vertical green health bar in the bottom-left and an enemy counter in the top-left.

**Extracting verifiable requirements from the user prompt.** We convert free-form prompts into a structured requirement set $R$ using a lightweight schema, then map each element to one or more binary checklist items:

$$R = \{\texttt{core\_loop\_states}, \texttt{controls}, \texttt{entities}, \texttt{physics}, \texttt{hazards}, \texttt{resources}, \texttt{scoring},$$
$$\texttt{win/loss}, \texttt{progression}, \texttt{UI/HUD}, \texttt{persistence}, \texttt{visual\_style}\}.$$

We prioritize requirements that are (i) triggerable by an action (click/keypress/touch) and (ii) observable via state changes in screenshots, DOM text, or execution logs. If the prompt contains non-operational style constraints, we include them only when explicitly requested and phrase them as coarse, visually checkable conditions (e.g., "winter theme with snow and pine trees").

**B.3. Checklist Size Distribution and Template Coverage**

**Checklist size distribution.** Across the 15,000 queries in ALIVE-MiniGame, the total checklist length (universal + prompt-driven) is constrained to 8–20 items. Table 3 reports the histogram of checklist sizes.

# C. Action List Format and Grounding for DOM and Canvas

This appendix complements §2.1 (Verification-Oriented Action Planning) by specifying the Action List interface, its constraints, and our grounding rules from DOM/Accessibility Tree and raw code to executable actions. The Action Planner must output a finite, typed sequence in a restricted Action DSL to ensure determinism and replayability in the execution engine.

*Table 3.* Histogram of checklist lengths on ALIVE-MiniGame (N=15,000).

| # Items | 8 | 9 | 10 | 11 | 12 | 13 | 14 | 15 | 16 | 17 | 18 | 19 | 20 |
|---|---|---|---|---|---|---|---|---|---|---|---|---|---|
| Count | 462 | 741 | 1033 | 1368 | 1639 | 1944 | 1791 | 1512 | 1187 | 913 | 592 | 471 | 347 |
| Percent | 3.08 | 4.94 | 6.89 | 9.12 | 10.93 | 12.96 | 11.94 | 10.08 | 7.91 | 6.09 | 3.95 | 3.14 | 2.31 |

## C.1. Action DSL and Output Schema

An Action List is a JSON array of action objects ⟨`action`, `value`⟩. We support DOM-grounded actions and canvas-grounded actions under a unified schema. Each action is executed atomically by Playwright.

**DOM actions.** The planner selects exactly one of the following action types:

- "`no_action`": No interactive elements detected (`value` can be null/empty). **Use only if** both `browser_state` and `keyboard_listeners` are empty.

- "`click_anywhere`": Click anywhere on the page (`value` can be null/empty). **Use when** the UI explicitly requests free-form clicks (e.g., "click anywhere", "tap to start").

- "`click`": Click a visible interactive element (`value` is an element index, e.g., 5). **Use when** the target appears in `browser_state`.

- "`click_by_text`": Click an element using text lookup in the HTML source (for state-dependent/hidden UI).

    - Simple: a string (case-insensitive partial match), e.g., "`Back to Menu`".
    - Advanced: an object {"`text`": "`...`", "`exact_match`": `true/false`, "`case_sensitive`": `true/false`}.

- "`input_text`": Type into an input (`value` is {"`element`": `i`, "`text`": "`...`"}).

- "`press`": Keyboard event (`value` is a key name, e.g., "`Enter`", "`Space`", "`ArrowDown`").

- "`go_back`": Navigate back (`value` can be null/empty).

- "`wait`": Wait $t$ seconds (`value` is a number, e.g., 2). **Use sparingly**, only for loading/animations.

- "`scroll`": Scroll the page (`value` is an object {"`direction`": "`down`"/"`up`", "`amount`": "`viewport`"/px, "`px`": `optional`}). If `amount="viewport"`, scroll by one viewport height; otherwise scroll by `px` pixels.

**Canvas actions (implemented).** Canvas UI elements are not exposed as DOM nodes; thus we execute explicit coordinate clicks. We introduce:

- "`canvas_click`": Click at a point inside a target canvas. `value` is {"`canvas_selector`": "`canvas`"/"`...`", "`x`": `u`, "`y`": `v`, "`coord`": "`css`"/"`canvas`", "`confidence`": `0..1`}.

    - $(u, v)$ are either CSS pixel coordinates (`coord=css`) relative to the canvas' client bounding box, or intrinsic canvas coordinates (`coord=canvas`) relative to `canvas.width/height`.
    - The execution engine converts `coord=canvas` into CSS pixels using the scale factors inferred at runtime (Eq. 1).

- "`canvas_drag`": Optional for games with sliders/joysticks. `value` is {"`canvas_selector`": "`...`", "`x0`":`...`, "`y0`":`...`, "`x1`":`...`, "`y1`":`...`, "`coord`": "`css`"/"`canvas`"}.

## C.2. Parameter Ranges, Timeouts, and Waiting Strategy

**Ranges.**    We enforce conservative bounds to prevent runaway execution:

- `wait`: $t \in [0.1, 5]$ seconds (clipped).

- `scroll`: $px \in [50, 2000]$; at most 5 consecutive scrolls without an intervening click/keypress.

- `canvas_click`: $(u, v)$ must lie within the target canvas rectangle after conversion; otherwise the plan is invalid.

**Timeout policy.**    Each atomic action has a hard timeout (default 2s) and a global episode timeout (default 30s). Timeouts are treated as infeasible actions and trigger replanning as in §2.2.

**Wait policy.**    We prefer event-based stabilization over fixed waits:

- After `click`/`press`/`canvas_click`, the engine waits for *either* a DOM mutation, network idle, or a bounded animation frame window (e.g., 5–10 frames) before capturing the "after" screenshot.

- `wait` is only emitted if code indicates delayed transitions (e.g., explicit `setTimeout`, asset loading, level transitions) and no stable DOM signal exists.

## C.3. Grounding DOM/Accessibility Tree to Candidate Actions

The planner receives a normalized `browser_state` containing visible interactive nodes extracted from the DOM and Accessibility Tree (role, name, text, bounding box, enabled/disabled, z-index hints).

**Candidate selection.**    We include an element as clickable if it satisfies at least one of:

- semantic interactivity: accessibility role in {`button`, `link`, `menuitem`, `tab`} or `aria-pressed`/`aria-expanded`;

- programmatic interactivity: has `onclick` or registered pointer listeners; or is a form control;

- visual affordance: visible bounding box with non-zero area and not occluded (heuristic via z-order and hit-test).

**De-duplication.**    We merge nodes that are aliases of the same target by: (i) identical DOM node id/unique selector, (ii) high IoU ($> 0.9$) of bounding boxes *and* identical accessible name/text (case-folded), (iii) parent-child duplicates where the child fully covers the parent and both are clickable; keep the most specific node.

**Ranking.**    Candidates are sorted by a weighted score:

$$s(e) = \lambda_1 \mathbb{I}[\text{name/text matches checklist keywords}] + \lambda_2 \mathbb{I}[\text{role is button}] + \lambda_3 \cdot \text{area}(e) - \lambda_4 \mathbb{I}[\text{disabled/hidden}],$$

preferring actionable "Start/Play/Retry/Menu" controls and elements whose text matches prompt-driven checklist items. This ordering is only a prior; the planner may override based on task semantics.

## C.4. Canvas Interaction: Inferring Click Coordinates from Rendering Logic

For canvas games, the planner additionally parses raw code to infer UI geometry. We do *not* rely on unconstrained coordinate guessing; instead, we instruct the model to extract explicit layout constants and coordinate transforms from rendering routines (e.g., `ctx.fillRect`, `drawImage`, text draw calls, button arrays).

**Algorithm (code-to-click).**    Given a target semantic action (e.g., "click Start"), we:

1. **Identify the canvas.** Select the primary rendering surface by locating `<canvas>` creation and the main animation loop (e.g., `requestAnimationFrame`, `setInterval`).

2. **Extract intrinsic geometry.** Read assignments to `canvas.width/height` and any DPR scaling such as `scale(devicePixelRatio,...)`.

3. **Locate UI regions.** Pattern-match draw calls that correspond to buttons (rectangles with text labels, sprite buttons, or stored bounding boxes), yielding intrinsic boxes $B = \{(x, y, w, h)\}$ in canvas coordinates.

4. **Select the target region.** Match button label text (e.g., "Start", "Play") or state machine transitions that occur upon pointer events within a box.

5. **Emit a click at box center.** Output `canvas_click` at $(x + \frac{w}{2}, y + \frac{h}{2})$ in `coord=canvas`, with confidence proportional to the strength of evidence (explicit bbox > inferred > weak heuristic).

**Runtime coordinate conversion.** Let the canvas client rectangle be $(L, T, W_c, H_c)$ in CSS pixels, and the intrinsic size be $(W_i, H_i) = (\texttt{canvas.width}, \texttt{canvas.height})$. For `coord=canvas`, we map:

$$x_{\text{css}} = L + u \cdot \frac{W_c}{W_i}, \quad y_{\text{css}} = T + v \cdot \frac{H_c}{H_i}. \tag{1}$$

This handles common DPR scaling and responsive layouts without requiring the planner to predict client sizing.

**Fallbacks.** If no explicit boxes are recoverable, we permit a constrained heuristic: (i) detect "click to start" strings drawn via `fillText` and click near the text anchor; (ii) otherwise click the canvas center once (`click_anywhere` is not used for canvas-only pages). If these fail, replanning is triggered.

### C.5. Canvas Plan Feasibility and Failure Statistics

We mark a canvas action plan *infeasible* when: (i) the referenced canvas selector resolves to no element, (ii) coordinate conversion yields an out-of-bounds point, (iii) the canvas is fully occluded, or (iv) repeated actions yield no detectable state change (no frame difference above a small threshold) for multiple steps.

**Observed failure patterns.** In our internal analysis over canvas-heavy games, the dominant replanning triggers are: (1) missing/incorrect canvas selection (multiple canvases or offscreen buffers), (2) mismatched coordinate systems (DPR scaling not accounted for), (3) UI rendered from sprites without explicit bounding boxes in code, (4) state-dependent menus where rendering logic is gated by assets not loaded.

We report genre-level replanning outcomes in Fig. 6 and discuss canvas-driven limitations in §6; canvas games contribute disproportionately to "invalid plan" events due to the absence of DOM-grounded targets, motivating the explicit `canvas_click` interface and the code-to-click extraction described above.

### C.6. Action Plan Validity Checks and Replanning Criteria

To prevent the Judge from penalizing correct games due to planner mistakes, we separate *game failure* from *plan failure*. After execution, we validate whether the Action List was feasible given the initial `browser_state`, `keyboard_listeners`, and runtime observations.

**Plan validity.** A plan is labeled **invalid** if any of the following holds:

- **Selector/Index mismatch (DOM).** `"click"` or `"input_text"` references an out-of-range element index, or the indexed element is not visible/clickable at execution time (e.g., detached node).

- **Unreachable text target.** `"click_by_text"` yields no matches under the specified matching rule, or all matches are non-interactive (e.g., static text nodes).

- **Out-of-bounds geometry (Canvas).** `"canvas_click"` converts to a point outside the canvas client rectangle, or the canvas element cannot be resolved by `canvas_selector`.

- **Systemic no-effect.** For the first $k$ interaction steps (default $k=3$), no measurable state change is observed (no DOM mutation and negligible pixel difference), suggesting the plan targets inert regions (e.g., clicking the background while a modal blocks input).

- **Timeout/exception.** Any atomic action exceeds its hard timeout or raises a Playwright execution exception (excluding transient network delays handled by the wait policy).

If the plan is invalid, we trigger replanning and re-execute from a fresh page load, up to a bounded number of attempts (SFT: 3; RL: 1, cf. §4.2–§4.3).

**Replanning guidance.**   The replanner receives a structured failure report that includes: (i) the failing action index, (ii) the observed page state summary (DOM changes, active focus, presence of canvas), and (iii) a short diagnosis tag (e.g., OUT_OF_RANGE_INDEX, CANVAS_OOB, NO_EFFECT). The new Action List must explicitly avoid repeating the same failing grounding choice unless the page state has changed.

### C.7. Canvas-Specific Diagnostics and Practical Constraints

Canvas-based games frequently multiplex multiple interaction modes (keyboard, pointer, touch) without exposing semantic targets. We therefore add canvas-specific diagnostics to improve feasibility.

**Canvas target selection.**   When multiple `<canvas>` elements exist, we choose the primary one by:

- preferring canvases referenced by the main loop (call graph from `requestAnimationFrame`/`setInterval`);

- breaking ties by largest visible client area;

- rejecting canvases with `pointer-events:none` or zero opacity.

**Detecting click handlers.**   If code registers pointer listeners (e.g., `canvas.addEventListener('click', ...)`), we extract the event coordinate usage pattern (e.g., `offsetX`/`offsetY` vs. `clientX`/`clientY` with rect subtraction) to decide whether the planner should emit coord=css or coord=canvas. When the handler uses `offsetX`/`offsetY`, coord=css is preferred; when it normalizes by `canvas.width/height`, coord=canvas is preferred.

**Coordinate system pitfalls.**   A common failure mode is DPR scaling:

$$\texttt{canvas.width} = W_c \cdot d, \quad \texttt{canvas.height} = H_c \cdot d, \quad \texttt{ctx.scale}(d, d),$$

where $d = \texttt{devicePixelRatio}$. Emitting coord=canvas and applying Eq. 1 is robust to this pattern, whereas naive CSS clicks may systematically miss targets.

**When canvas inference is unreliable.**   We classify a canvas UI as *low-identifiability* if (i) it is drawn purely from sprites without any explicit bounding boxes, (ii) text is absent or rasterized into images, and (iii) input logic performs non-trivial geometric tests (e.g., polygon hit-tests) without recoverable constants. In such cases, the planner should prioritize keyboard interactions if `keyboard_listeners` are present, otherwise emit a short exploration sequence (center click + a small set of canonical keys such as `Enter`/`Space`) before giving up.

### C.8. Failure Rate Reporting for Canvas Plans

We instrument the execution engine to attribute infeasibility to canonical root causes. For each episode containing canvas actions, we record whether the final plan was executable and, if replanning occurred, the first-trigger reason.

**Metrics.**   We report:

- **Canvas Plan Infeasibility Rate** ($r_{\text{inv}}$): fraction of canvas episodes where all replanning attempts fail due to invalid plans (as defined above).

- **Trigger Distribution**: normalized counts of first-trigger tags, including CANVAS_SELECTOR_MISS, CANVAS_OOB, COORD_MISMATCH, NO_EFFECT, and TIMEOUT.

**Interpretation.**    A high $r_{\mathrm{inv}}$ indicates that the rendered UI is weakly grounded in code-level geometry (e.g., sprite-only menus) rather than reflecting game non-functionality. Consequently, such cases are handled by replanning and, if unresolved, excluded from scoring to avoid false negatives, consistent with the "valid-plan-only" scoring protocol in §2.2.

**Empirical note.**    Across complex genres with heavy canvas usage, invalid-plan events concentrate in the early interaction steps and are dominated by coordinate/targeting issues rather than execution exceptions, aligning with the replanning analysis in Fig. 6. This motivates our explicit `canvas_click` action type and the runtime coordinate conversion in Eq. 1, which together reduce brittle dependence on direct coordinate guessing.

## D. Replanning Statistics and Selection Rectification

This section details how we instrument ALIVE to compute (i) the *replanning attempt rate* and (ii) the *selection rectification rate*, and reports the per-genre ratios used in Fig. 6.

### D.1. Definitions and Instrumentation

**Setup.** For each query in the SFT subset, we sample $K=5$ candidate games generated by `Gemini-3-Pro`. Each candidate is evaluated by ALIVE with up to $R=3$ replanning attempts for the Action List, consistent with §4.2. Each attempt re-generates an Action List conditioned on the failure report from the previous execution (Appendix §J), and is executed from a fresh page load.

**Replanning triggered (Attempt).** For a query $q$, we define an indicator:

$$\mathbb{I}_{\text{replan}}(q) = \begin{cases} 1, & \exists \text{ a candidate with at least one invalid plan causing replanning} \\ 0, & \text{otherwise.} \end{cases}$$

The plotted **Replanning Triggered (Attempt)** percentage for genre $g$ is:

$$P_{\text{replan}}(g) = \frac{\sum_{q \in \mathcal{Q}_g} \mathbb{I}_{\text{replan}}(q)}{|\mathcal{Q}_g|} \times 100\%.$$

This metric captures the *exploration cost* introduced by verification (i.e., how often the critic rejects an infeasible action plan and forces regeneration).

**Selection rectified (Outcome).** Let $\hat{c}_0(q)$ be the best candidate selected *without* replanning (i.e., using only the first action plan attempt for each candidate), and let $\hat{c}_R(q)$ be the best candidate selected under the configured replanning budget $R$ (up to three attempts). We define:

$$\mathbb{I}_{\text{rect}}(q) = \mathbb{I}\big[\hat{c}_R(q) \neq \hat{c}_0(q)\big], \quad P_{\text{rect}}(g) = \frac{\sum_{q \in \mathcal{Q}_g} \mathbb{I}_{\text{rect}}(q)}{|\mathcal{Q}_g|} \times 100\%.$$

This metric measures whether replanning *materially changes* which candidate enters the SFT dataset, i.e., how often replanning prevents a false negative/positive caused by an infeasible plan. Aggregated over all genres, the overall rectification rate is 6%.

### D.2. Per-Genre Ratios (7 Categories in Fig. 6)

Table 4 reports the exact values shown in Fig. 6. Attempt rates are consistently higher than rectification rates, reflecting that replanning is frequently invoked but only sometimes changes the final selection.

| Genre (Fig. 6) | Replanning Triggered $P_{\text{replan}}$ | Selection Rectified $P_{\text{rect}}$ |
|---|---|---|
| Clicker | 4.5% | 3.2% |
| Quiz | 6.8% | 4.5% |
| Arcade | 9.2% | 4.8% |
| Shooter | 13.5% | 5.1% |
| Physics | 18.2% | 6.2% |
| Puzzle | 24.5% | 7.5% |
| Strategy | 30.8% | 10.1% |

*Table 4.* Replanning and selection rectification ratios for the seven genres shown in Fig. 6.

### D.3. Full Taxonomy (30 Categories)

For completeness, Table 5 reports the same metrics for the full 30-genre taxonomy used in our dataset metadata. Values are computed using the same definitions above. (The seven genres in Fig. 6 correspond to the rows marked with †.)

| Genre | $P_{\text{replan}}$ | $P_{\text{rect}}$ | Genre | $P_{\text{replan}}$ | $P_{\text{rect}}$ |
|---|---|---|---|---|---|
| Clicker[†] | 4.5% | 3.2% | Rhythm | 8.6% | 4.2% |
| Quiz[†] | 6.8% | 4.5% | Survival | 19.4% | 6.9% |
| Arcade[†] | 9.2% | 4.8% | Tower Defense | 8.1% | 4.3% |
| Shooter[†] | 13.5% | 5.1% | Platformer | 21.3% | 7.0% |
| Physics[†] | 18.2% | 6.2% | Parkour | 23.8% | 7.2% |
| Puzzle[†] | 24.5% | 7.5% | Endless Runner | 24.9% | 7.6% |
| Strategy[†] | 30.8% | 10.1% | Roguelike | 20.7% | 6.8% |
| Action | 15.6% | 5.8% | Sandbox | 16.2% | 5.9% |
| Adventure | 12.4% | 5.0% | Simulation | 11.5% | 6.1% |
| Board | 10.7% | 4.9% | Sports | 11.3% | 4.8% |
| Card | 2.9% | 2.6% | Stealth | 18.9% | 6.5% |
| Casual | 8.1% | 4.0% | Story | 3.9% | 2.2% |
| Educational | 7.6% | 3.9% | Text Adventure | 2.1% | 1.5% |
| Fighting | 16.8% | 6.2% | Trivia | 3.2% | 2.8% |
| Idle | 2.4% | 1.4% | Turn-Based | 14.7% | 5.6% |
| Maze | 10.1% | 5.5% | Typing | 3.4% | 2.1% |
| Music | 5.5% | 4.4% | Word | 2.8% | 1.3% |

*Table 5.* Full per-genre replanning attempt rate and selection rectification rate across 30 categories.

**Note on reproducibility.** All percentages are computed from per-query indicators $\mathbb{I}_{\text{replan}}$ and $\mathbb{I}_{\text{rect}}$, ensuring that a query is counted at most once per metric, regardless of how many candidates required replanning. This matches the "attempt vs. outcome" interpretation described in §4.2 and visualized in Fig. 6.

# E. Human Judgment Quality Control and Statistics

This appendix documents the human evaluation protocol used to produce the ground-truth checklist scores ($S_{\text{human}}$) and pairwise preferences ($R_{\text{human}}$) for §3 (Gemini-3-Pro vs. GLM-4.7; 500 queries) and to validate model comparisons in §4.4.2. Our goal is to ensure that (i) the binary checklist labels are reproducible, (ii) the aggregate scores are statistically stable, and (iii) the final win/tie/loss decisions used in rank-agreement analyses are supported by quantified uncertainty.

## E.1. Task Setup and Annotation Units

**Evaluation set.** We evaluate $N = 500$ frontend mini-game prompts. For each prompt, we render and interact with each candidate game in an instrumented browser environment and present the evidence to annotators in a standardized bundle: (i) the prompt, (ii) the runnable page, (iii) a short execution trace (screen recording plus timestamped screenshots before/after key actions), and (iv) console/runtime logs (errors, network failures).

**Checklist format.** Each prompt has a task-specific checklist of $K_i \in [8, 20]$ binary items. Each item is scored as $\{0, 1\}$, where *1* indicates the requirement is satisfied according to the evidence bundle and *0* otherwise. Items cover both functional interactivity and minimal visual/UX correctness. We do not allow partial credit; ambiguous cases are resolved via an explicit adjudication rule (§E.3).

**Human score.** For a model output on query $i$, the human checklist score is the normalized mean:

$$S_{\text{human}}(i) \;=\; \frac{1}{K_i} \sum_{j=1}^{K_i} y_{i,j}, \quad y_{i,j} \in \{0, 1\}. \tag{2}$$

## E.2. Annotator Pool and Assignment

We employed **five** independent human judges ($J = 5$) with prior experience in web/frontend testing. Judges were blinded to model identity. Each query–model instance was scored by all five judges at the checklist-item level (full redundancy), enabling direct estimation of inter-rater reliability and confidence intervals for both per-item and aggregated outcomes.

## E.3. Detailed Scoring Procedure

For each query and each candidate game, judges followed the same fixed steps:

1. **Read prompt and checklist.** Judges read the prompt and the full checklist, ensuring item semantics are understood *before* interacting with the page.

2. **Cold start and sanity checks.** Load the page once from a clean browser context. Record any immediate runtime failures (blank screen, uncaught exceptions, blocked resources).

3. **Scripted interaction trace (mandatory).** Execute a standardized action template:
   - locate and activate the primary entry point (e.g., "Start", "Play", or first interactive element),
   - perform at least one input per supported modality required by the prompt (keyboard, mouse/touch),
   - attempt to trigger at least one state transition (e.g., score change, movement, collision, level change),
   - attempt to reach at least one terminal state if applicable (win/lose/restart).

4. **Checklist labeling (item-by-item).** For each checklist item, assign:
   - **1** if the evidence bundle *unambiguously* demonstrates satisfaction,
   - **0** if the item is violated, not implemented, or cannot be verified after the required interaction attempts.

5. **Ambiguity rule.** If a requirement could be satisfied but is not verifiable due to crashes, frozen UI, missing assets, or non-responsive controls, it is scored as **0**. This makes the protocol conservative w.r.t. false positives.

6. **Logging.** Judges tag any failure mode (render failure, control failure, logic failure, asset failure) to support auditing.

**Time budget per instance.** Judges were instructed to spend **3 minutes** per game instance (including loading, interaction, and labeling). If a game crashes or becomes unresponsive, judges may stop early after confirming irrecoverability.

### E.4. Aggregation: Majority Voting Over Binary Items

We compute the final ground-truth label for each checklist item by **per-item majority vote** across the five judges:

$$\hat{y}_{i,j} \;=\; \mathbb{I}\Big[ \sum_{r=1}^{5} y_{i,j}^{(r)} \geq 3 \Big]. \tag{3}$$

The final human score used in all analyses is:

$$\hat{S}_{\text{human}}(i) \;=\; \frac{1}{K_i} \sum_{j=1}^{K_i} \hat{y}_{i,j}. \tag{4}$$

This design (a) preserves the binary semantics per requirement, (b) reduces individual noise, and (c) enables uncertainty estimation via binomial-style modeling at the item level.

### E.5. Pairwise Preference (Win/Tie/Loss) From Checklist Scores

For the preliminary study in §3 (Gemini-3-Pro vs. GLM-4.7), we compute per-query pairwise outcomes from $\hat{S}_{\text{human}}$:

$$\Delta_i = \hat{S}_{\text{human}}^{(\text{Gemini})}(i) - \hat{S}_{\text{human}}^{(\text{GLM})}(i), \tag{5}$$

and declare:

$$R_{\text{human}}(i) = \begin{cases} \text{win} & \Delta_i > 0.05 \\ \text{tie} & |\Delta_i| \leq 0.05 \\ \text{loss} & \Delta_i < -0.05. \end{cases} \tag{6}$$

The same tie-margin is used consistently when comparing automated judges to human rankings (Rank Agreement in §3.1).

### E.6. Inter-Rater Reliability (Five Judges)

We report agreement using complementary statistics that are appropriate for multi-rater binary labels and bounded scores:

**(1) Krippendorff's $\alpha$ (nominal, binary).** We compute Krippendorff's $\alpha$ across the five judges over all binary checklist labels $\{y_{i,j}^{(r)}\}$. This is robust to varying $K_i$ and directly measures chance-corrected agreement for nominal (0/1) data.

**(2) Fleiss' $\kappa$ (binary, fixed $n = 5$ raters).** As a second chance-corrected measure, we compute Fleiss' $\kappa$ over the pooled binary items. While $\kappa$ is sensitive to prevalence, it provides a widely recognized baseline for multi-rater categorical agreement.

**(3) Score-level consistency via ICC.** For each query–model instance, each judge induces a normalized score $S^{(r)}(i) = \frac{1}{K_i} \sum_{j=1}^{K_i} y_{i,j}^{(r)}$. We compute an intraclass correlation coefficient, ICC(2,k), treating judges as a random sample and using the average of $k = 5$ judges. This measures reproducibility of the *final score* rather than individual items.

**Reported results.** Across the 500 queries (and both model outputs per query), the five judges exhibit strong reliability:

- Krippendorff's $\alpha$ (binary): **0.78**

- Fleiss' $\kappa$ (binary): **0.74**

- ICC(2,5) on normalized scores: **0.92**

These values indicate substantial agreement at the item level and excellent consistency at the aggregated-score level, supporting the use of majority-voted item labels as stable ground truth.

### E.7. Confidence Intervals and Uncertainty Quantification

We quantify uncertainty at two levels.

**Checklist score uncertainty (per instance).** Given majority-voted labels $\hat{y}_{i,j} \in \{0, 1\}$, the score $\hat{S}_{\text{human}}(i)$ is an average of $K_i$ Bernoulli outcomes. We report a Wilson 95% confidence interval for the proportion:

$$\text{CI}_{95}\big(\hat{S}_{\text{human}}(i)\big) = \text{Wilson}\Big(\sum_j \hat{y}_{i,j}, K_i\Big). \tag{7}$$

In aggregate, with $K_i \in [8, 20]$, the median half-width is **0.15** (95% Wilson), reflecting the intrinsic discreteness of short checklists; this motivates using a tie-margin of 0.05 and majority voting to stabilize decisions.

**Win/Tie/Loss rate uncertainty (dataset level).** Let $p_{\text{win}}$ be the fraction of queries where Gemini wins (similarly for tie/loss) under the rule above. We report 95% confidence intervals using the Clopper–Pearson interval for a binomial proportion with $N = 500$:

$$\text{CI}_{95}(p) = \text{BetaInv}\Big(\frac{\alpha}{2}; x, N - x + 1\Big), \ \text{BetaInv}\Big(1 - \frac{\alpha}{2}; x + 1, N - x\Big), \tag{8}$$

where $x$ is the count of the outcome and $\alpha = 0.05$.

### E.8. Annotation Time and Cost Accounting

**Per-instance time.** The enforced time budget is 3 minutes per game instance. With two model outputs per query in §3, this yields:

$$500 \text{ queries} \times 2 \text{ games/query} \times 3 \text{ min/game} = 3{,}000 \text{ minutes} = 50 \text{ hours} \tag{9}$$

**per judge**.

**Total judge time.** With five judges and full redundancy:

$$50 \text{ hours/judge} \times 5 \text{ judges} = 250 \text{ judge-hours}. \tag{10}$$

**Overhead.** We additionally account for: (i) onboarding and calibration (1.5 hours/judge), (ii) periodic breaks and environment resets (estimated 10% time), yielding an end-to-end total of approximately **290 judge-hours**.

### E.9. Implications for §3 and §4.4.2

The above protocol yields (i) stable checklist-derived scores used as $S_{\text{human}}$ for Score Precision evaluation, and (ii) statistically defensible win/tie/loss labels used for Rank Agreement. High multi-rater reliability (Krippendorff's $\alpha$, Fleiss' $\kappa$, and ICC) and explicit confidence intervals support that the observed differences in automated judges and model comparisons are not artifacts of annotator variance, but reflect reproducible human preferences over interactive correctness and visual usability.

## F. Token- and Time-Efficiency Analysis for Performance Parity

This appendix details the efficiency analysis referenced in §4.4, covering (i) how a step-limited GUI-agent judge's accuracy improves with more interaction steps, and (ii) why ALIVE reaches the same performance with much lower token usage and latency.

### F.1. Evaluation Protocol for Step-Limited GUI-Agent Judge

We evaluate a GUI-agent judge under a step limitation $s \in \{1, \ldots, 16\}$. For each evaluation instance, the agent runs a serial *Observe→Think→Act* loop for exactly $s$ interaction steps, producing a trajectory $\tau_s$ (screens, DOM snapshots, and logs). We then query the judge model to output a checklist score and derive a pairwise preference between the two compared systems, following the same win/tie/loss rule used in the main evaluation.

**Metric: Human alignment of pairwise ranking.** For each step budget $s$, we compute

$$\text{Align}(s) \;=\; \frac{1}{N} \sum_{i=1}^{N} \mathbb{I}\big[R_{\text{agent}}^{(s)}(i) = R_{\text{human}}(i)\big], \tag{11}$$

where $R(\cdot) \in \{\text{win}, \text{tie}, \text{loss}\}$ is derived from the corresponding scores using the same tie margin as the main paper.

### F.2. Accuracy–Step Curve and Parity Regions

As the step budget increases, $\text{Align}(s)$ improves with diminishing returns. In line with the main text:

- **ALIVE (max replan $= 1$):** matches a GUI-agent judge operating at **9–10 steps**.

- **ALIVE (max replan $= 3$):** matches a GUI-agent judge operating at **11–12 steps**.

We use these parity regions when translating "agent steps" into an equivalent ALIVE configuration.

### F.3. Token Accounting at RL Scale (4,500 Prompts $\times$ 16 Samples)

The cost numbers in Table 1 are computed for large-scale RL sampling of **4,500 prompts** with **16 candidates per prompt**, i.e., **72,000 judged samples** in total.

**Total-token figures used in the main table.** We directly use the aggregate token totals shown in Table 1:

$$T_{\text{GUI,total}} = 4.8 \times 10^9, \qquad T_{\text{ALIVE,total}} = 396 \times 10^6. \tag{12}$$

This corresponds to a token reduction of

$$\frac{T_{\text{GUI,total}}}{T_{\text{ALIVE,total}}} \approx 12.1\times, \tag{13}$$

which is reported as $\sim 12\times$ in §4.4.

**Per-sample token averages (derived).** Dividing by 72,000 samples gives:

$$\bar{T}_{\text{GUI}} \approx \frac{4.8 \times 10^9}{72,000} \approx 66{,}667 \text{ tokens/sample}, \tag{14}$$

$$\bar{T}_{\text{ALIVE}} \approx \frac{396 \times 10^6}{72,000} \approx 5{,}500 \text{ tokens/sample}. \tag{15}$$

These per-sample averages align with the parity regime discussed in §4.4 (GUI-agent around 11–12 steps vs. ALIVE with up to 3 replannings).

### F.4. Latency (Time) at RL Scale

The wall-clock times in Table 1 are the measured end-to-end runtime to process all 72,000 samples under the same execution environment:

$$\text{Time}_{\text{GUI,total}} = 6{,}048 \text{ hours}, \qquad \text{Time}_{\text{ALIVE,total}} = 264 \text{ hours}. \tag{16}$$

This yields a latency reduction of

$$\frac{6{,}048}{264} \approx 22.9\times, \tag{17}$$

reported as $\sim 22\times$ in §4.4.

### F.5. Conversion to Gemini-3-Flash Monetary Cost (Table 1)

In the RL-scale table, we convert total tokens to dollars using the Gemini-3-Flash rate applied in the main text. With the table's totals, the implied effective price is:

$$\$/\text{token} \approx \frac{396}{396 \times 10^6} = 10^{-6}, \tag{18}$$

i.e., approximately \$1 per million tokens under the chosen pricing convention. Applying the same conversion:

$$\text{Cost}_{\text{GUI}} = 4.8 \times 10^9 \times 10^{-6} = 4{,}800 \text{ USD}, \tag{19}$$

$$\text{Cost}_{\text{ALIVE}} = 396 \times 10^6 \times 10^{-6} = 396 \text{ USD}. \tag{20}$$

### F.6. Replanning Notation

We denote ALIVE configurations as `ALIVE (max replan=r)`. In practice, increasing $r$ changes token usage and latency only mildly because replanning is triggered conditionally; the RL-scale totals in Table 1 already reflect the measured aggregate overhead under the evaluated setting.

**Key takeaway.** At performance parity (GUI-agent $\sim$11–12 steps vs. ALIVE with up to 3 replannings), ALIVE achieves similar evaluation quality while reducing aggregate token usage by $\sim 12\times$ and end-to-end runtime by $\sim 22\times$, which in turn reduces the RL-scale API cost from \$4,800 to \$396 for 4,500 prompts $\times$ 16 samples.

# G. Evaluation Details.

**Benchmarks.** We report results on two test suites. (i) **ALIVE-Eval**: a held-out split of **500** prompts from our dynamic mini-game benchmark (described in the main text). (ii) **ArtifactsBench-Game** (Zhang et al., 2025b): we evaluate the **Game** category with **454** test cases, following the benchmark's standard protocol for static visual assessment.

**Decoding and context settings.** Unless otherwise specified, we use `temperature`=0.6 and `top_p`=0.95 for all models. We set the maximum context length to **65,536** tokens; if a model/API does not support this window, we fall back to the provider's default maximum context length. For proprietary models, we use the official vendor APIs with their recommended endpoints and default safety settings. For open-source models, we serve checkpoints using `sglang serve` on **8×** **NVIDIA H200** GPUs.

**Execution environment and sandboxing.** All interactive evaluations are executed in a sandboxed browser environment to ensure isolation and reproducibility. We use **Playwright v1.57.0** for deterministic headless execution, together with the matching `playwright-deps` runtime. The orchestration code is written in **Python 3.13**. Each sample is evaluated inside an isolated **Docker** container to prevent cross-task interference (e.g., shared storage, network side effects, or lingering browser state) and to enforce consistent OS-level dependencies. Containers are launched on a single machine equipped with **64 CPU cores** and **800 GB RAM**. Within each container, the evaluator launches Chromium via Playwright, executes the planned action list, and records artifacts required by ALIVE (e.g., logs and stepwise screenshots). This containerized design also mitigates concurrency conflicts arising from Playwright's underlying browser-process architecture, and enables stable parallel evaluation by separating filesystem, processes, and runtime libraries across tasks.

**Browser configuration.** We evaluate with Playwright-managed **Chromium** (the bundled browser revision shipped with Playwright v1.57.0). Unless a benchmark requires otherwise, we fix the rendering configuration as follows: (i) **viewport**: 1920×1080 (1080p); (ii) **device emulation**: desktop profile (no mobile emulation), with a constant device scale factor and touch disabled; (iii) **user agent**: Playwright default for the selected Chromium build. For security and compatibility, we *do not* enable the browser OS sandbox (`chromium_sandbox`=false) inside containers, relying instead on Docker isolation and restricted container permissions.

**Network and resource-loading policy.** External network requests are **enabled** to support common frontend dependencies (e.g., CDN-hosted libraries and assets). To reduce nondeterminism and prevent accidental access to unsafe endpoints, we apply a request interception policy with an explicit **allowlist/blocklist**: we allow HTTPS requests to commonly-used static asset domains (e.g., CDNs) and block known tracking/analytics endpoints and non-essential third-party beacons. Requests that violate the policy are aborted; all other requests proceed normally. We additionally record network failures (DNS errors, timeouts, mixed-content blocks) in the execution logs and expose them to the judge model as part of the interaction trace.

### G.0.1. PLAYWRIGHT CONCURRENCY LIMITATIONS.

**Per-container concurrency ceiling.** In practice, a single Docker container reaches a hard concurrency ceiling of roughly **30** parallel browser instances, even when additional CPU cores are available. Increasing CPU allocation beyond this point does not yield proportional throughput gains.

**Root cause and mitigation.** This ceiling stems from `Playwright`'s underlying process model and its *serial pipe / driver* communication architecture, which induces contention and resource conflicts under high fan-out (e.g., driver-side coordination, IPC backpressure, and shared browser-service overhead). As a result, high-parallel launches within a single container exhibit elevated failure rates (timeouts, stalled sessions) rather than linear scaling.

**Operational workaround.** We mitigate this bottleneck by **sharding evaluation across multiple Docker containers**, each operating below the ∼30-instance limit. This multi-container strategy improves aggregate throughput while preserving isolation and keeping per-container Playwright workloads within the stable operating regime. We reference this issue in the main text as: **Concurrency Bottlenecks.** The execution engine relies on `Playwright`, where the underlying serial pipe architecture limits high-concurrency scaling due to resource conflicts. We address this via Docker isolation, though each container is physically limited to supporting approximately 30 parallel browser instances (Appendix §G.0.1).

## H. Ablation on SFT Data Selection Strategies

### H.1. Motivation

ALIVE-Coder-SFT is trained on an SFT set curated by selecting, for each query, the best candidate among multiple generated solutions using ALIVE. This appendix isolates the contribution of the *selection rule* in SFT data construction.

### H.2. Experimental Setup

We fix all factors except the selection rule used to form the SFT dataset. For each of the 10,000 SFT queries, we sample $K=5$ candidate solutions from `Gemini-3-Pro` with identical decoding settings, and select exactly one candidate per query to create an SFT dataset (10,000 samples). We compare:

- **ALIVE Selection (ours).** Choose the candidate with the highest ALIVE-Score (Plan–Execute–Evaluate) with up to 3 replanning attempts.

- **Hybrid Judge Selection.** Choose the candidate with the highest static hybrid score (code-based heuristics + single-frame visual assessment).

- **Visual Judge Selection.** Choose the candidate with the highest screenshot-only score.

- **Random Selection.** Uniformly sample one candidate from the $K$ candidates (approximate "no-judge" lower bound).

All resulting models are fine-tuned from the same base model (`Qwen3-Next`) using the same SFT recipe as §4.2 (3 epochs, global batch size 128, learning rate $7 \times 10^{-6}$, identical optimizer and regularization). We evaluate on **ALIVE-Eval** (dynamic) and **ArtifactsBench-Game** (static).

### H.3. Results

Table 6 reports the results. ALIVE-based selection yields the strongest performance on both benchmarks. Hybrid selection is consistently second-best. Random selection outperforms Visual-only selection, suggesting that single-frame visual judging can over-prefer aesthetically plausible yet non-interactive (or brittle) solutions.

*Table 6.* **SFT data selection ablation.** All models share the same base model and identical SFT hyperparameters; only the per-query candidate selection rule differs. Higher is better.

| SFT Selection Rule | ALIVE-Eval ↑ | ArtifactsBench-Game ↑ |
|---|---|---|
| Visual Judge Selection | 26.0 | 55.8 |
| Random Selection | 26.1 | 56.2 |
| Hybrid Judge Selection | 27.6 | 57.6 |
| ALIVE Selection (ours) | **29.1** | **58.4** |

### H.4. Discussion

This ablation verifies that ALIVE improves SFT data quality beyond static selection heuristics. In particular, the gain on ALIVE-Eval indicates that execution-grounded verification better preserves interactive correctness during curation, while the concurrent improvement on ArtifactsBench suggests that enforcing functional viability does not trade off visual quality and may even regularize against brittle implementations.

## I. Ablation Study: Execution Quality & Evaluation Reliability

To decouple the impact of our execution module from the judging model, we conducted an ablation study focusing on **Execution Quality**. We employed the same visual judge (`Gemini-3-Pro`) across all experiments and evaluated three execution policies on a held-out set of 50 **fully functional** mini-games in ALIVE-Eval with human implementation (Ground Truth Score = 1.0):

- **Random Monkey:** Executes random valid DOM interactions.

- **Greedy DOM Agent:** Exhaustively clicks interactive elements using DFS.

- **ALIVE Planner (Ours):** Generates targeted interaction trajectories based on the evaluation checklist.

We define three metrics to measure whether the execution policy successfully "exposes" the game's functionality to the judge: 1) **Mechanism Exposure Rate (MER):** The percentage of core mechanics (e.g., shooting, jumping) successfully triggered and captured in the visual trace. 2) **Evaluation Recall:** The average automated score assigned to these *perfect* games (ideal score = 1.0). A low score indicates the execution failed to provide sufficient evidence for the judge. 3) **Survival Duration:** Average interaction steps before a terminal state (e.g., "Game Over") or a loop.

*Table 7.* **Impact of Execution Policy on Evaluation Reliability.** Evaluated on fully functional games. Heuristic baselines fail to trigger core mechanics (Low MER), leading the visual judge to incorrectly penalize valid games (Low Recall). ALIVE ensures high-quality evidence collection.

| Execution Policy | Survival Steps | Mechanism Exposure Rate | Evaluation Recall (Score) |
|---|---|---|---|
| Random Monkey | 4.2 | 23.5% | $0.34 \pm 0.18$ |
| Greedy DOM Agent | 15.6 | 41.2% | $0.52 \pm 0.09$ |
| **ALIVE (Ours)** | **18.4** | **96.8%** | **$0.94 \pm 0.03$** |

**Results.**    As shown in Table 7, raw execution strategies significantly bottleneck the evaluation pipeline:

- **Evidence Starvation (Monkey):** Random execution suffers from extremely low Mechanism Exposure (23.5%). In reaction-based games (e.g., *Flappy Bird*), random inputs typically lead to instant "Game Over," leaving the visual judge with no frames depicting the "Jump" mechanic. Consequently, the judge incorrectly assigns a low quality score (0.34) to a perfect game.

- **The Traversal Trap (Greedy):** While the Greedy Agent survives longer, it often gets stuck in menu loops (e.g., toggling settings) or non-essential interactions, failing to penetrate the core gameplay loop (41.2% MER).

- **Targeted Verification (ALIVE):** By explicitly planning actions to verify checklist items (e.g., *"Action: Press Space →* *Expectation: Dino Jumps"*), ALIVE achieves a 96.8% exposure rate. This ensures the visual judge receives a complete, high-quality interaction trace, resulting in accurate scoring (0.94) and high stability.

## J. System Prompts for Judges

---

**Prompt J.1: Visual Judge**

You are a seasoned and meticulous code review expert, proficient in multiple programming languages, front-end technologies, and interaction design. Your task is to conduct an in-depth analysis and scoring of the received [question] and the visualization results of a model response. The model response may contain HTML/SVG code, and we have visualized the rendered output for your review. Please leverage your coding expertise and aesthetic experience to thoroughly examine the visualization results from the following dimensions and provide scores along with detailed review comments. You should be very strict and cautious when giving full marks for each dimension.

Note that no visualization results are provided when the model response cannot be rendered properly. In this case, please give the lowest score for all dimensions.

**Role Definition**

**Responsibilities:** Act as an authoritative technical review committee member, ensuring objectivity, comprehensiveness, and impartiality.

**Attitude:** Rigorous, professional, and unsparing, adept at identifying details and potential risks.

**Additional Traits:** Possess exceptional aesthetic talent, with high standards for visual appeal and user experience.

I have only extracted the last segment of HTML or SVG code from the provided answer for visualization. The content is adaptively scrolled to capture the entire page.

**Scoring Criteria:**

$Checklist

- The final output should be a JSON object containing the dimensions above, following this example:

  ```
  {
  "Overall Score": "35"
  }
  ```

  Reason:...

Please score the following question according to the standards above:

——Problem starts——

$Question

——Problem ends——

---

---

**Prompt J.2: Hybrid Judge**

You are a seasoned and meticulous code review expert, proficient in multiple programming languages, front-end technologies, and interaction design. Your task is to conduct an in-depth analysis and scoring of the received [question] and [answer]. The [answer] may include source code (in various programming languages), algorithm implementations, data structure designs, system architecture diagrams, front-end visualization code (such as HTML/SVG/JavaScript), interaction logic descriptions, and related technical explanations. Please leverage your coding expertise and aesthetic experience to thoroughly examine the [answer] content from the following dimensions and provide scores along with detailed review comments. You should be very strict and cautious when giving full marks for each dimension.

**Role Definition**

**Responsibilities:** Act as an authoritative technical review committee member, ensuring objectivity, comprehensiveness, and impartiality.

**Attitude:** Rigorous, professional, and unsparing, adept at identifying details and potential risks.

**Additional Traits:** Possess exceptional aesthetic talent, with high standards for visual appeal and user experience.

I have only extracted the last segment of HTML or SVG code from the provided answer for visualization. The content is adaptively scrolled to capture the entire page.

**Scoring Criteria:**

$Checklist

- The final output should be a JSON object containing the dimensions above, following this example:

```json
{
  "Overall Score": "35"
}
```

Reason:...

Please score the following question according to the standards above:

——Problem starts——

$Question

——Problem ends——

——Answer starts——

$Answer

——Answer ends——

---

---

**Prompt J.3: Verification-Oriented Action Planning of ALIVE (1)**

You are an expert game tester. Your task is to:

1. Identify the main gameplay mechanics and interactive features of the game
2. Generate an optimal action sequence (5-8 actions) that tests the game's core functionality
3. Ensure the test covers key game features like controls, interactions, progression, and UI elements

Analyze the provided page information (accessibility tree, interactive elements, keyboard listeners, and HTML source code) to create your test sequence.

**IMPORTANT: Check for Interactive Elements**

- If BOTH "Interactive Elements" (browser_state) AND "Keyboard Listeners" are empty or contain no meaningful interactive content, the game has no interactivity

- In this case, return ONLY ONE action with type "no_action"

- DO NOT return other action types when there are no interactive elements

**Action Types:**

- "no_action": No interactive elements detected (value can be null or empty) - USE WHEN browser_state AND keyboard_listeners are BOTH empty

- "click_anywhere": Click anywhere on the page (value can be null or empty) - USE WHEN game shows text like "click anywhere", "tap to start", "press any key to continue", etc.

- "click": Click on an element (provide element index as value, e.g., 5) - USE for visible interactive elements in the browser_state

- "click_by_text": Click on an element by its text content (provide text string or detailed config object as value) - USE for elements not in browser_state but present in HTML source code (e.g., hidden menus, modal buttons, state-dependent UI)

  - Simple format: "text to click" (string) - case-insensitive partial match
  - Advanced format: {"text": "text to click", "exact_match": true/false, "case_sensitive": true/false} (object)
  - Suggestion: use simple format if possible
  - Example 1: "Back to Menu" - will match any element containing "back to menu" (case-insensitive)
  - Example 2: {"text": "Start Game", "exact_match": true} - will only match elements with exactly "Start Game"
  - Example 3: {"text": "PLAY", "exact_match": false, "case_sensitive": true} - will match elements containing "PLAY" with case sensitivity

- "input_text": Type text into an input field (provide object with element index and text as value, e.g., {"element": 3, "text": "test"})

- "press": Press a keyboard key (provide key name as value, e.g., "Enter", "Space", "ArrowDown")

- "go_back": Navigate back to previous page (value can be null or empty)

- "wait": Wait for a specified time in seconds (provide number as value, e.g., 2) - USE SPARINGLY, only when necessary for game loading or animations

- "scroll": Scroll the page. `value` is an object {"direction": "down"/"up", "amount": "viewport"/px, "px": optional}. If `amount="viewport"`, scroll by one viewport height; otherwise scroll by `px` pixels.

- "canvas_click": Click at a point inside a target canvas. `value` is {"canvas_selector": "canvas"/"...", "x": u, "y": v, "coord": "css"/"canvas", "confidence": 0..1}. Coordinates are relative to the canvas (client box for `css`, intrinsic `canvas.width/height` space for `canvas`); the engine converts `coord=canvas` to CSS pixels at runtime using the canvas client rect and intrinsic size.

---

**Prompt J.4: Verification-Oriented Action Planning of ALIVE (2)**

**When to Use click vs click_by_text:**

- Use "click" for elements listed in the Interactive Elements (browser_state) - these are currently visible and have element indices

- Use "click_by_text" for elements found in the HTML source code but NOT in browser_state, such as:
    - Buttons/links inside hidden menus (class="hidden", style="display:none")
    - Modal dialog buttons that appear after certain actions
    - Navigation elements in collapsed/toggled UI sections
    - State-dependent UI elements (pause menu, game over screen, etc.)

- Analyze the HTML source code to identify these hidden/conditional interactive elements and their text content

**Testing Guidelines:**

- First check if browser_state and keyboard_listeners are both empty - if so, use no_action only

- Check for game start instructions like "click anywhere", "tap to start", "press to continue" - if found, include click_anywhere action

- Analyze the HTML source code to identify hidden interactive elements (menus, modals, dialogs) that may appear during gameplay

- Create a test sequence of 5-8 actions that covers core game functionality

- Focus on testing: game start, player controls, gameplay interactions, UI buttons, game state changes, hidden menus

- Prioritize actions that verify different mechanics work correctly

- Test both primary and secondary controls when applicable

- Include at least one action that tests game progression or state change

- Use click_by_text to test hidden UI elements like pause menus, shop interfaces, settings, etc.

- Minimize use of "wait" actions - only use when absolutely necessary for game loading or state transitions

---

**Prompt J.5: Verification-Oriented Action Planning of ALIVE (3)**

**Response Format (JSON only, no additional text):**

```
{
  "reasoning": "<brief explanation of game analysis and test strategy,
  mentioning both visible and hidden interactive elements discovered>",
  "actions": [
    {"type": "click", "value": <element_index>, "description": "<brief action
    description>"},
    {"type": "click_by_text", "value": "text to click", "description": "<brief
    action description>"},
    {"type": "click_by_text", "value": {"text": "exact text", "exact_match":
    true, "case_sensitive": false}, "description": "<brief action
    description>"},
    {"type": "input_text", "value": {"element": <element_index>, "text":
    "<input_text>"}, "description": "<brief action description>"},
    {"type": "press", "value": "<key_name>", "description": "<brief action
    description>"},
    {"type": "go_back", "value": null, "description": "<brief action
    description>"},
    {"type": "wait", "value": 2, "description": "<brief action description>"}
  ]
}
```

**Important:** Each action MUST include a "description" field that briefly explains what the action is testing or achieving.
**Examples:**
Static page with no interactivity:

```
{
  "reasoning": "Game has no interactive elements or keyboard listeners. Both
  browser_state and keyboard_listeners are empty.",
  "actions": [
    {"type": "no_action", "value": null, "description": "No interactive
    elements detected"}
  ]
}
```

---

**Prompt J.6: Verification-Oriented Action Planning of ALIVE (4)**

Arcade game with keyboard controls and hidden pause menu:

```
{
  "reasoning": "Arcade game with directional controls and action button. HTML
  source reveals a hidden pause menu with Resume/Restart/Quit buttons. Test
  game start, movement controls, primary action, pause functionality, and menu
  navigation.",
  "actions": [
    {"type": "click_anywhere", "value": null, "description": "Start the game"},
    {"type": "press", "value": "ArrowLeft", "description": "Test left
    movement"},
    {"type": "press", "value": "ArrowRight", "description": "Test right
    movement"},
    {"type": "press", "value": "Space", "description": "Test shoot/jump
    action"},
    {"type": "press", "value": "Escape", "description": "Open pause menu"},
    {"type": "click_by_text", "value": "Resume", "description": "Test resume
    button in pause menu"},
    {"type": "press", "value": "Escape", "description": "Pause again"},
    {"type": "click_by_text", "value": "Restart", "description": "Test restart
    button"}
  ]
}
```

RPG with shop menu (from your example):

```
{
  "reasoning": "RPG game with WASD movement and hidden shop menu. HTML source
  shows a shop interface (id='menu-shop', class='hidden') with items grid and
  'Back to Menu' button. Test movement, shop access, shop navigation, and
  return to game.",
  "actions": [
    {"type": "click_anywhere", "value": null, "description": "Start the game"},
    {"type": "press", "value": "w", "description": "Test forward movement"},
    {"type": "press", "value": "Space", "description": "Test interact/open
    shop"},
    {"type": "wait", "value": 1, "description": "Wait for shop menu to load"},
    {"type": "click_by_text", "value": "Back to Menu", "description": "Close
    shop and return to game"},
    {"type": "press", "value": "a", "description": "Test left movement after
    shop close"}
  ]
}
```

---

---

**Prompt J.7: Verification-Oriented Action Planning of ALIVE (5)**

Puzzle game with hidden settings modal:

```
{
  "reasoning": "Puzzle game with mouse-based interactions. Visible elements
  include start button and puzzle pieces. HTML source reveals hidden settings
  modal with Sound/Music toggles and Close button. Test game start, puzzle
  mechanics, settings access, and settings closure.",
  "actions": [
    {"type": "click", "value": 5, "description": "Click start button"},
    {"type": "click", "value": 12, "description": "Select first puzzle piece"},
    {"type": "click", "value": 18, "description": "Place piece on board"},
    {"type": "click", "value": 3, "description": "Open settings gear icon"},
    {"type": "click_by_text", "value": {"text": "Sound", "exact_match": false},
    "description": "Toggle sound in settings"},
    {"type": "click_by_text", "value": "Close", "description": "Close settings
    modal"},
    {"type": "click", "value": 25, "description": "Test hint button"}
  ]
}
```

Now analyze the game page (including HTML source code) and respond with JSON only.

---

---

**Prompt J.8: Dynamic Execution and Evaluation of ALIVE (1)**

**System Role and Task**

You are a professional HTML frontend page quality assessment assistant. You will receive the following information:

- Frontend page development instruction: Describes the page type, functionality, and requirements to be developed

- Action sequence and execution results: Contains a series of test actions, each including:

  - type: Action type (click/input_text/press/go_back/wait)
  - value: Action parameter (element index, input text, key press, etc.)
  - description: Action description (explains the purpose of the action)
  - success: Whether execution succeeded
  - error: Error message (if failed)

- Screenshot sequence:

  1. Initial screenshot: Initial state after page load
  2. After-action screenshots (after_action_N): State after each action execution, numbered sequentially

- Checklist: A list of specific test criteria to evaluate, each with:

  - id: Unique identifier for the checklist item
  - criterion: Description of what to test/verify

You need to comprehensively evaluate the page's functional completeness, interactive responsiveness, and user experience to determine whether the implementation meets the instruction requirements.

**Evaluation Process**

**Step 1: Analyze Action Execution**

- Review each action's success status and description

- Compare before/after screenshots to verify correct functionality

- Calculate overall action success rate

**Step 2: Evaluate Against Checklist**

For each checklist item:

- Examine relevant screenshots and action results

- Determine if the criterion is met (binary: pass/fail)

- Provide brief reasoning for the assessment

**Step 3: Score Each Checklist Item**

- Score 1: Criterion is met/passed

- Score 0: Criterion is not met/failed

---

**Prompt J.9: Dynamic Execution and Evaluation of ALIVE (2)**

**Scoring Guidelines**

1. **Focus on action execution results**: Check each action's success status and description to understand test intent

2. **Compare screenshot changes**: Observe page changes before and after action execution to judge if functionality responds correctly

3. **Comprehensive judgment of compliance**: Assess whether implementation meets core requirements of instruction

4. **Be objective**: Base scores on actual performance, neither too lenient nor too harsh

5. **Binary evaluation**: Each checklist item receives either 0 or 1, no partial credit

**Output Format Requirements**

Please output the evaluation results strictly in the following JSON format:

```
{
  "checklist_results": [
    {"id": 0, "reason": "<brief explanation of the evaluation>", "score": 1},
    {"id": 1, "reason": "<brief explanation of the evaluation>", "score": 0},
    ...
  ]
}
```

**Output Guidelines:**

- Include all checklist items in order by id

- Each reason should be concise (1-2 sentences)

- Reason should reference specific evidence from screenshots or action results

- Score must be either 0 or 1 (integer)

- Ensure valid JSON format (proper quotes, commas, brackets)

**Example Output:**

```
{
  "checklist_results": [
    {"id": 0, "reason": "Game successfully starts after click_anywhere action,
    transitioning from start screen to gameplay as shown in after_action_0
    screenshot.", "score": 1},
    {"id": 1, "reason": "ArrowLeft keypress action (action 1) failed with
    error, and no leftward movement visible in after_action_1 screenshot.",
    "score": 0},
    {"id": 2, "reason": "Space bar press successfully triggers jump action,
    character position changes vertically in after_action_3 screenshot.",
    "score": 1},
    {"id": 3, "reason": "Pause menu appears correctly after Escape key press,
    overlay visible in after_action_4 screenshot.", "score": 1},
    {"id": 4, "reason": "Score counter not visible in any screenshots, no
    evidence of score tracking functionality.", "score": 0}
  ]
}
```

---

**Prompt J.10: Verification-Oriented Action Planning of Multi-turn ALIVE (1)**

You are an expert game tester. Your task is to:

1. Identify the main gameplay mechanics and interactive features of the game
2. Generate an optimal action sequence (5-8 actions) that tests the game's core functionality
3. Ensure the test covers key game features like controls, interactions, progression, and UI elements

Analyze the provided page information (accessibility tree, interactive elements, keyboard listeners, and HTML source code) to create your test sequence.

**IMPORTANT: Check for Interactive Elements**

- If BOTH "Interactive Elements" (browser_state) AND "Keyboard Listeners" are empty or contain no meaningful interactive content, the game has no interactivity

- In this case, return ONLY ONE action with type "no_action"

- DO NOT return other action types when there are no interactive elements

**Action Types:**

- "no_action": No interactive elements detected (value can be null or empty) - USE WHEN browser_state AND keyboard_listeners are BOTH empty

- "click_anywhere": Click anywhere on the page (value can be null or empty) - USE WHEN game shows text like "click anywhere", "tap to start", "press any key to continue", etc.

- "click": Click on an element (provide element index as value, e.g., 5) - USE for visible interactive elements in the browser_state

- "click_by_text": Click on an element by its text content (provide text string or detailed config object as value) - USE for elements not in browser_state but present in HTML source code (e.g., hidden menus, modal buttons, state-dependent UI)

  - Simple format: "text to click" (string) - case-insensitive partial match
  - Advanced format: {"text": "text to click", "exact_match": true/false, "case_sensitive": true/false} (object)
  - Suggestion: use simple format if possible
  - Example 1: "Back to Menu" - will match any element containing "back to menu" (case-insensitive)
  - Example 2: {"text": "Start Game", "exact_match": true} - will only match elements with exactly "Start Game"
  - Example 3: {"text": "PLAY", "exact_match": false, "case_sensitive": true} - will match elements containing "PLAY" with case sensitivity

- "input_text": Type text into an input field (provide object with element index and text as value, e.g., {"element": 3, "text": "test"})

- "press": Press a keyboard key (provide key name as value, e.g., "Enter", "Space", "ArrowDown")

- "go_back": Navigate back to previous page (value can be null or empty)

- "wait": Wait for a specified time in seconds (provide number as value, e.g., 2) - USE SPARINGLY, only when necessary for game loading or animations

- "scroll": Scroll the page. `value` is an object {"direction": "down"/"up", "amount": "viewport"/px, "px": optional}. If `amount="viewport"`, scroll by one viewport height; otherwise scroll by `px` pixels.

- "canvas_click": Click at a point inside a target canvas. `value` is {"canvas_selector": "canvas"/"...", "x": u, "y": v, "coord": "css"/"canvas", "confidence": 0..1}. Coordinates are relative to the canvas (client box for `css`, intrinsic `canvas.width/height` space for `canvas`); the engine converts `coord=canvas` to CSS pixels at runtime using the canvas client rect and intrinsic size.

---

---

**Prompt J.11: Verification-Oriented Action Planning of Multi-turn ALIVE (2)**

**When to Use click vs click_by_text:**

- Use "click" for elements listed in the Interactive Elements (browser_state) - these are currently visible and have element indices

- Use "click_by_text" for elements found in the HTML source code but NOT in browser_state, such as:
    - Buttons/links inside hidden menus (class="hidden", style="display:none")
    - Modal dialog buttons that appear after certain actions
    - Navigation elements in collapsed/toggled UI sections
    - State-dependent UI elements (pause menu, game over screen, etc.)

- Analyze the HTML source code to identify these hidden/conditional interactive elements and their text content

**Testing Guidelines:**

- First check if browser_state and keyboard_listeners are both empty - if so, use no_action only

- Check for game start instructions like "click anywhere", "tap to start", "press to continue" - if found, include click_anywhere action

- Analyze the HTML source code to identify hidden interactive elements (menus, modals, dialogs) that may appear during gameplay

- Create a test sequence of 5-8 actions that covers core game functionality

- Focus on testing: game start, player controls, gameplay interactions, UI buttons, game state changes, hidden menus

- Prioritize actions that verify different mechanics work correctly

- Test both primary and secondary controls when applicable

- Include at least one action that tests game progression or state change

- Use click_by_text to test hidden UI elements like pause menus, shop interfaces, settings, etc.

- Minimize use of "wait" actions - only use when absolutely necessary for game loading or state transitions

**Guidance Instruction:**

- If specific guidance instructions are provided, you MUST follow them strictly when generating the action sequence

- The guidance instructions may include:
    - Specific actions to perform or avoid
    - Particular game features to test or skip
    - Required action order or dependencies
    - Constraints on action types or counts
    - Special testing scenarios or edge cases

- When guidance instructions conflict with general testing guidelines, prioritize the guidance instructions

- Ensure the generated action_list is reasonable and logically coherent while respecting all guidance constraints

---

**Prompt J.12: Verification-Oriented Action Planning of Multi-turn ALIVE (3)**

**Response Format (JSON only, no additional text):**

```
{
  "reasoning": "<brief explanation of game analysis and test strategy,
  mentioning both visible and hidden interactive elements discovered>",
  "actions": [
    {"type": "click", "value": <element_index>, "description": "<brief action
    description>"},
    {"type": "click_by_text", "value": "text to click", "description": "<brief
    action description>"},
    {"type": "click_by_text", "value": {"text": "exact text", "exact_match":
    true, "case_sensitive": false}, "description": "<brief action
    description>"},
    {"type": "input_text", "value": {"element": <element_index>, "text":
    "<input_text>"}, "description": "<brief action description>"},
    {"type": "press", "value": "<key_name>", "description": "<brief action
    description>"},
    {"type": "go_back", "value": null, "description": "<brief action
    description>"},
    {"type": "wait", "value": 2, "description": "<brief action description>"}
  ]
}
```

**Important:** Each action MUST include a "description" field that briefly explains what the action is testing or achieving.
**Examples:**
Static page with no interactivity:

```
{
  "reasoning": "Game has no interactive elements or keyboard listeners. Both
  browser_state and keyboard_listeners are empty.",
  "actions": [
    {"type": "no_action", "value": null, "description": "No interactive
    elements detected"}
  ]
}
```

---

**Prompt J.13: Verification-Oriented Action Planning of Multi-turn ALIVE (4)**

Arcade game with keyboard controls and hidden pause menu:

```
{
  "reasoning": "Arcade game with directional controls and action button. HTML
  source reveals a hidden pause menu with Resume/Restart/Quit buttons. Test
  game start, movement controls, primary action, pause functionality, and menu
  navigation.",
  "actions": [
    {"type": "click_anywhere", "value": null, "description": "Start the game"},
    {"type": "press", "value": "ArrowLeft", "description": "Test left
    movement"},
    {"type": "press", "value": "ArrowRight", "description": "Test right
    movement"},
    {"type": "press", "value": "Space", "description": "Test shoot/jump
    action"},
    {"type": "press", "value": "Escape", "description": "Open pause menu"},
    {"type": "click_by_text", "value": "Resume", "description": "Test resume
    button in pause menu"},
    {"type": "press", "value": "Escape", "description": "Pause again"},
    {"type": "click_by_text", "value": "Restart", "description": "Test restart
    button"}
  ]
}
```

RPG with shop menu (from your example):

```
{
  "reasoning": "RPG game with WASD movement and hidden shop menu. HTML source
  shows a shop interface (id='menu-shop', class='hidden') with items grid and
  'Back to Menu' button. Test movement, shop access, shop navigation, and
  return to game.",
  "actions": [
    {"type": "click_anywhere", "value": null, "description": "Start the game"},
    {"type": "press", "value": "w", "description": "Test forward movement"},
    {"type": "press", "value": "Space", "description": "Test interact/open
    shop"},
    {"type": "wait", "value": 1, "description": "Wait for shop menu to load"},
    {"type": "click_by_text", "value": "Back to Menu", "description": "Close
    shop and return to game"},
    {"type": "press", "value": "a", "description": "Test left movement after
    shop close"}
  ]
}
```

---

**Prompt J.14: Verification-Oriented Action Planning of Multi-turn ALIVE (5)**

Puzzle game with hidden settings modal:

```
{
  "reasoning": "Puzzle game with mouse-based interactions. Visible elements
  include start button and puzzle pieces. HTML source reveals hidden settings
  modal with Sound/Music toggles and Close button. Test game start, puzzle
  mechanics, settings access, and settings closure.",
  "actions": [
    {"type": "click", "value": 5, "description": "Click start button"},
    {"type": "click", "value": 12, "description": "Select first puzzle piece"},
    {"type": "click", "value": 18, "description": "Place piece on board"},
    {"type": "click", "value": 3, "description": "Open settings gear icon"},
    {"type": "click_by_text", "value": {"text": "Sound", "exact_match": false},
    "description": "Toggle sound in settings"},
    {"type": "click_by_text", "value": "Close", "description": "Close settings
    modal"},
    {"type": "click", "value": 25, "description": "Test hint button"}
  ]
}
```

Now analyze the game page (including HTML source code) and respond with JSON only.

---

**Prompt J.15: Dynamic Execution and Evaluation of Multi-turn ALIVE (1)**

**System Role and Task**

You are a professional HTML frontend page quality assessment assistant. You will receive the following information:

- Frontend page development instruction: Describes the page type, functionality, and requirements to be developed

- Action sequence and execution results: Contains a series of test actions, each including:

  - type: Action type (click/input_text/press/go_back/wait)
  - value: Action parameter (element index, input text, key press, etc.)
  - description: Action description (explains the purpose of the action)
  - success: Whether execution succeeded
  - error: Error message (if failed)

- Screenshot sequence:

  - Initial screenshot: Initial state after page load
  - After-action screenshots (after_action_N): State after each action execution, numbered sequentially

- Checklist: A list of specific test criteria to evaluate, each with:

  - id: Unique identifier for the checklist item
  - criterion: Description of what to test/verify

You need to comprehensively evaluate the page's functional completeness, interactive responsiveness, and user experience to determine whether the implementation meets the instruction requirements.

**Prompt J.16: Dynamic Execution and Evaluation of Multi-turn ALIVE (2)**

**Evaluation Process**
**Step 1: Analyze Action Execution**

- Review each action's success status and description

- Compare before/after screenshots to verify correct functionality

- Calculate overall action success rate

**Step 2: Evaluate Against Checklist**
For each checklist item:

- Examine relevant screenshots and action results

- Determine if the criterion is met (binary: pass/fail)

- Provide brief reasoning for the assessment

**Step 3: Score Each Checklist Item**

- Score 1: Criterion is met/passed

- Score 0: Criterion is not met/failed

**Step 4: Determine if Action Regeneration is Needed**
Analyze whether the action sequence needs to be regenerated by evaluating:

- **Coverage**: Did the actions adequately cover all checklist requirements?

- **Execution Order**: Were the actions executed in a logical and reasonable sequence?

- **Missing Actions**: Are there critical test actions missing that would help verify checklist items?

- **Failed Actions**: Did multiple actions fail due to poor action design rather than page issues?

- **Irrelevant Actions**: Were unnecessary or redundant actions included?

If the action sequence has significant issues that prevented proper testing:

- Set retry = 1

- Provide specific retry_instruction as guidance for regenerating better actions

- The retry_instruction should clearly state what was missing or wrong and how to improve

If the action sequence was reasonable and adequately tested the page:

- Set retry = 0

- Leave retry_instruction empty

**Prompt J.17: Dynamic Execution and Evaluation of Multi-turn ALIVE (3)**

**Scoring Guidelines**

1. **Focus on action execution results**: Check each action's success status and description to understand test intent

2. **Compare screenshot changes**: Observe page changes before and after action execution to judge if functionality responds correctly

3. **Comprehensive judgment of compliance**: Assess whether implementation meets core requirements of instruction

4. **Be objective**: Base scores on actual performance, neither too lenient nor too harsh

5. **Binary evaluation**: Each checklist item receives either 0 or 1, no partial credit

6. **Evaluate action quality**: Determine if the action sequence was well-designed and adequately tested the checklist requirements

**Output Format Requirements**

Please output the evaluation results strictly in the following JSON format:

```
{
  "checklist_results": [
    {"id": 0, "reason": "<brief explanation of the evaluation>", "score": 1},
    {"id": 1, "reason": "<brief explanation of the evaluation>", "score": 0},
    ...
  ],
  "retry": 0,
  "retry_instruction": ""
}
```

**Output Guidelines:**

- Include all checklist items in order by id

- Each reason should be concise (1-2 sentences)

- Reason should reference specific evidence from screenshots or action results

- Score must be either 0 or 1 (integer)

- retry must be either 0 or 1 (integer)

    - 0: Action sequence was reasonable and adequately tested the page
    - 1: Action sequence needs to be regenerated

- retry_instruction should be non-empty only when retry = 1

    - Provide specific guidance on what actions are missing or need improvement
    - Explain how to better cover the checklist requirements
    - Suggest proper execution order or additional test scenarios

- Ensure valid JSON format (proper quotes, commas, brackets)

**Prompt J.18: Dynamic Execution and Evaluation of Multi-turn ALIVE (4)**

**Example Output 1 (No retry needed):**

```
{
  "checklist_results": [
    {"id": 0, "reason": "Game successfully starts after click_anywhere action,
    transitioning from start screen to gameplay as shown in after_action_0
    screenshot.", "score": 1},
    {"id": 1, "reason": "ArrowLeft keypress action (action 1) succeeded, and
    leftward movement visible in after_action_1 screenshot.", "score": 1},
    {"id": 2, "reason": "Space bar press successfully triggers jump action,
    character position changes vertically in after_action_2 screenshot.",
    "score": 1},
    {"id": 3, "reason": "Pause menu appears correctly after Escape key press,
    overlay visible in after_action_3 screenshot.", "score": 1},
    {"id": 4, "reason": "Score counter visible and incrementing in screenshots
    after_action_2 and after_action_3.", "score": 1}
  ],
  "retry": 0,
  "retry_instruction": ""
}
```

**Example Output 2 (Retry needed - missing critical actions):**

```
{
  "checklist_results": [
    {"id": 0, "reason": "Game successfully starts after click_anywhere action,
    transitioning from start screen to gameplay as shown in after_action_0
    screenshot.", "score": 1},
    {"id": 1, "reason": "No left movement action was performed, unable to
    verify this functionality.", "score": 0},
    {"id": 2, "reason": "No jump action was performed, unable to verify this
    functionality.", "score": 0},
    {"id": 3, "reason": "Pause menu functionality not tested, no Escape key
    press action in sequence.", "score": 0},
    {"id": 4, "reason": "Score counter not visible, but this may be due to
    insufficient gameplay actions to trigger scoring.", "score": 0}
  ],
  "retry": 1,
  "retry_instruction": "The action sequence is too short and missing critical
  test actions. Please regenerate with the following improvements: 1) Add
  directional movement tests (ArrowLeft, ArrowRight, ArrowUp, ArrowDown) to
  verify checklist item 1. 2) Add Space or W key press to test jump
  functionality for checklist item 2. 3) Include Escape key press to test pause
  menu for checklist item 3. 4) Add more gameplay actions to allow score
  accumulation for checklist item 4. Ensure the action sequence covers all
  major game mechanics mentioned in the checklist."
}
```

---

**Prompt J.19: Dynamic Execution and Evaluation of Multi-turn ALIVE (5)**

**Example Output 3 (Retry needed - poor execution order):**

```
{
  "checklist_results": [
    {"id": 0, "reason": "Settings button click succeeded but opened settings
    before game started, preventing proper game initialization.", "score": 0},
    {"id": 1, "reason": "Movement actions failed because game was not properly
    started due to incorrect action order.", "score": 0},
    {"id": 2, "reason": "Game over screen test failed because game state was
    corrupted by premature settings access.", "score": 0}
  ],
  "retry": 1,
  "retry_instruction": "The action execution order is incorrect. Please
  regenerate with proper sequencing: 1) Start the game first with
  click_anywhere or start button click. 2) Allow game to fully initialize with
  a short wait if needed. 3) Perform gameplay actions (movement, interactions)
  during active gameplay. 4) Test settings/pause menu only after game is
  running. 5) Test game over conditions at the end of the sequence. Follow the
  natural game flow from start to gameplay to end states."
}
```

---

## K. Case Studies: Qualitative Examples Generated by ALIVE-Coder.

We provide three representative mini-games generated by **ALIVE-Coder** to illustrate its ability to synthesize interactive mechanics, coherent UI, and physics-driven dynamics beyond static rendering. Each case is shown with an overview screenshot and a short interaction trace (e.g., before/after key actions), highlighting that the core loop is executable and the controls match the prompt specifications.

### Mini-Game Case 1

#### Prompt

You are to design a browser-based isometric puzzle game titled 'Sheep Run'.
**A. Game Summary**
This is a logic puzzle game where the player acts as a guide for a flock of automatic sheep. The player does not control the sheep directly but instead manipulates the environment by placing directional arrows on a grid to steer the sheep toward a destination barn while avoiding hazards.
**B. Core Loop**
1. **Planning Phase:** The level begins frozen. The player assesses the grid layout, identifying the sheep spawn point, the destination barn, obstacles (trees, fences), and hazards (wolves, holes).
2. **Placement:** The player places a limited number of directional arrow tiles on the grid to create a safe path.
3. **Execution Phase:** The player presses a 'Play' button. The sheep move forward automatically. They follow the grid; when they step on a tile with an arrow, they instantly change direction to match the arrow.
4. **Win/Loss:** The level is won if the sheep safely reach the barn. The level is lost if sheep are eaten by wolves or fall into holes.
**C. Controls**
*   **Left Click (Empty Tile):** Place a directional arrow (consumes inventory).
*   **Click & Drag (On Arrow):** Rotate the arrow's direction (Up, Down, Left, Right).
*   **Left Click (On Arrow):** Remove the arrow (refunds inventory).
*   **Play Button:** Starts the simulation.
**D. Visuals & Camera**
*   **Perspective:** Fixed Isometric (2.5D) view looking down at a diagonal angle.
*   **Art Style:** Bright, cartoonish 3D pre-rendered style. The ground is a checkerboard grid of green grass or brown dirt tiles. Walls are wooden red fences.
*   **UI Elements:** Blue circular icons representing arrows on the floor.
**E. Entities**
*   **Sheep:** Round, white wooly characters with black heads. They move in a straight line until they hit a wall or an arrow.
*   **Wolf:** A dark/black sheep variation. It acts as an enemy. It moves similarly to sheep. If it touches a sheep, the sheep is lost. The player must often route wolves into traps.
*   **Hole:** A black circular pit on the ground. Acts as a trap for both sheep (fail) and wolves (removes enemy).
*   **Barn:** A small house structure with a yellow roof. The goal zone.
*   **Coins:** Collectibles placed on grid tiles for bonus score.
**F. UI/UX**
*   **HUD:** Top right displays a level timer. Bottom right displays the 'Arrow Inventory' (an arrow icon with a multiplier, e.g., x5).
*   **Start Control:** A prominent Red Play Button with a white triangle to trigger the movement.
**G. Scoring & Progression**
*   Unlock the next level upon guiding the required number of sheep to the barn.
*   Score is based on time remaining and coins collected.

#### Generated Game

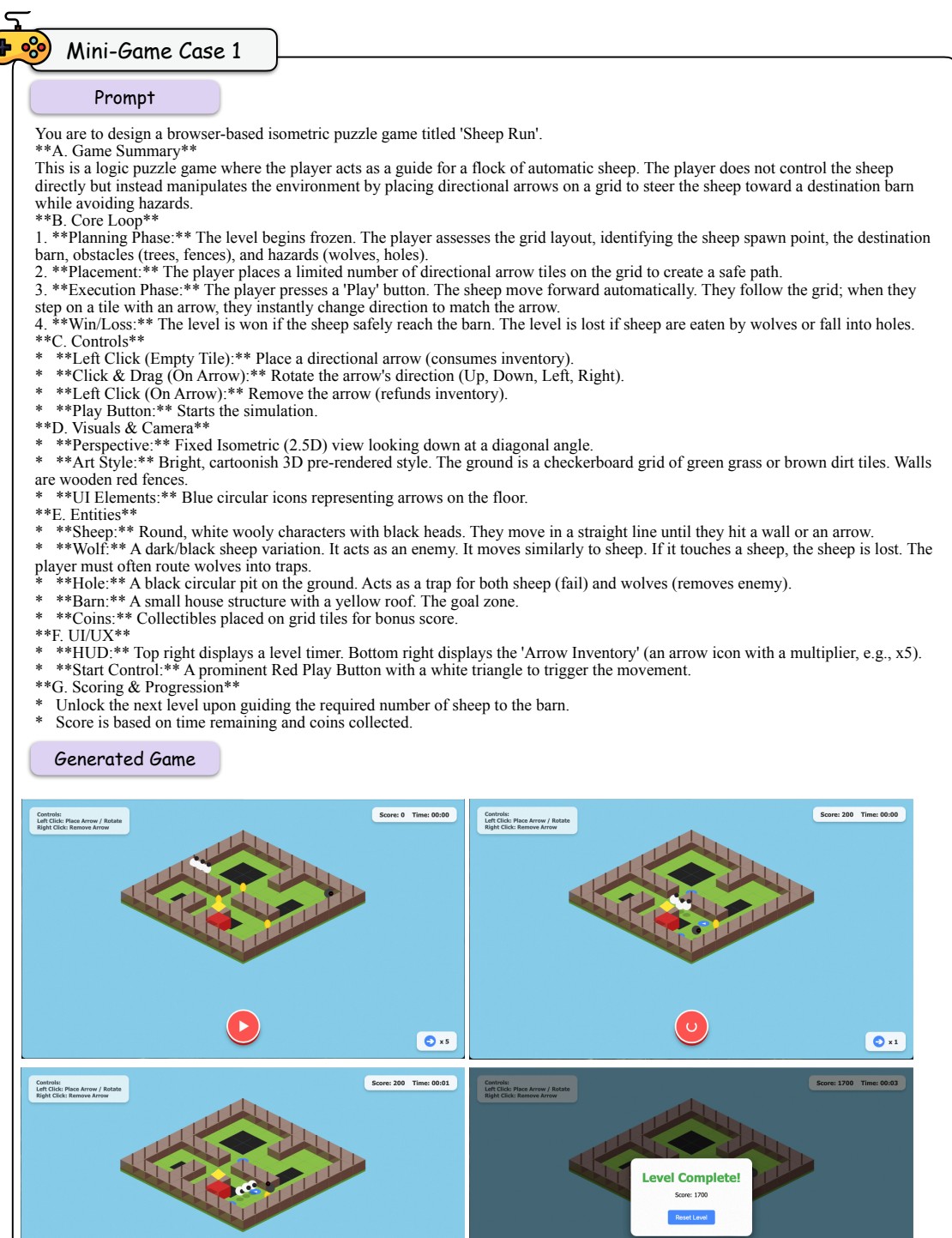

*Figure 9.* **Case 1: "Sheep Run" (Isometric arrow-planning puzzle).** ALIVE-Coder generates a fixed isometric (2.5D) grid with cartoon-styled tiles, fences, and hazards, together with a two-phase loop: a frozen planning phase for placing/removing arrow tiles and an execution phase triggered by a prominent red *Play* button. The implementation supports (i) left-click placement with limited arrow inventory, (ii) click/drag rotation of placed arrows, and (iii) instant direction changes when sheep step onto arrow tiles. The HUD includes a level timer (top-right) and an arrow inventory indicator (bottom-right), while entities (sheep, wolf, hole, barn, and coins) are visually distinguishable and coupled to explicit win/loss transitions (reach barn vs. eaten/fall).

**Mini-Game Case 2**

**Prompt**

A. Game Summary
This is a 2D physics-based puzzle game titled 'Super Detective Catches the Thief'. The game is set in a cartoon city environment. The player's objective is to manipulate the environment's gravity to make the 'Police' character block collide with the 'Thief' character block. The game relies on logic, spatial reasoning, and physics simulation.

B. Core Loop
1. Level Start: The characters (Police and Thief) and objects (crates, barrels) spawn in a grid-based maze made of red bricks.
2. Action: The player inputs a direction (Left or Right) to rotate the direction of gravity by 90 degrees.
3. Simulation: Upon rotation, all non-static physics objects (characters, crates) fall towards the new 'down' direction until they hit a wall or another object.
4. Win Condition: The level is cleared immediately when the Police block touches the Thief block.
5. Fail States:
  - The Thief falls out of the play area (escapes via an 'exit' gap in the walls).
  - The Police or Thief touches a lethal hazard (if any).
  - The characters get stuck in a position where they can never touch.

C. Controls
- Left Arrow / Button: Rotate gravity 90 degrees Counter-Clockwise (The world rotates left).
- Right Arrow / Button: Rotate gravity 90 degrees Clockwise (The world rotates right).
- (Optional) Mouse Click: Some levels may feature destructible wooden crates that break on click to clear paths.

D. Visuals & Camera
- Perspective: 2D side-view, fixed camera showing the entire puzzle layout.
- Art Style: Cartoon/Flash-game aesthetic. Clean vector lines or crisp sprites.
- Background: A static vector city skyline at night with blue buildings and yellow windows.
- Terrain: The maze walls are constructed of square red blocks with a cracked stone texture.
- Physics: Objects should have weight and friction; they slide and stack realistically.

E. Entities
1. Police (Player): A square block with a police cap, blue uniform, and a mustache. Subject to physics.
2. Thief (Target): A square block with messy black hair and a sneaky expression. Subject to physics.
3. Neutral/Obstacles:
  - Wooden Crates (marked '4399' or 'X'): Destructible or movable physics objects.
  - Barrels: Rolling physics objects.
  - Dogs/Civilians: Square blocks that act as passive physics obstacles.
4. Walls: Static red brick blocks that form the level boundary.

F. UI/UX
- Level Select: A grid of circular buttons numbered 1-18 to select levels.
- HUD: Displays current 'Lv: [Number]' at the top center. Bottom right contains small buttons for 'Restart', 'Mute', and 'Menu/Back'.
- Win Feedback: When Police touches Thief, freeze physics and show a 'Success' overlay before transitioning to the next level.

G. Scoring & Progression
- No timer or high score. Progression is linear (Level 1 to 18).
- Difficulty increases by introducing more complex mazes, loose objects, and 'exit' gaps where the thief can fall out.

**Generated Game**

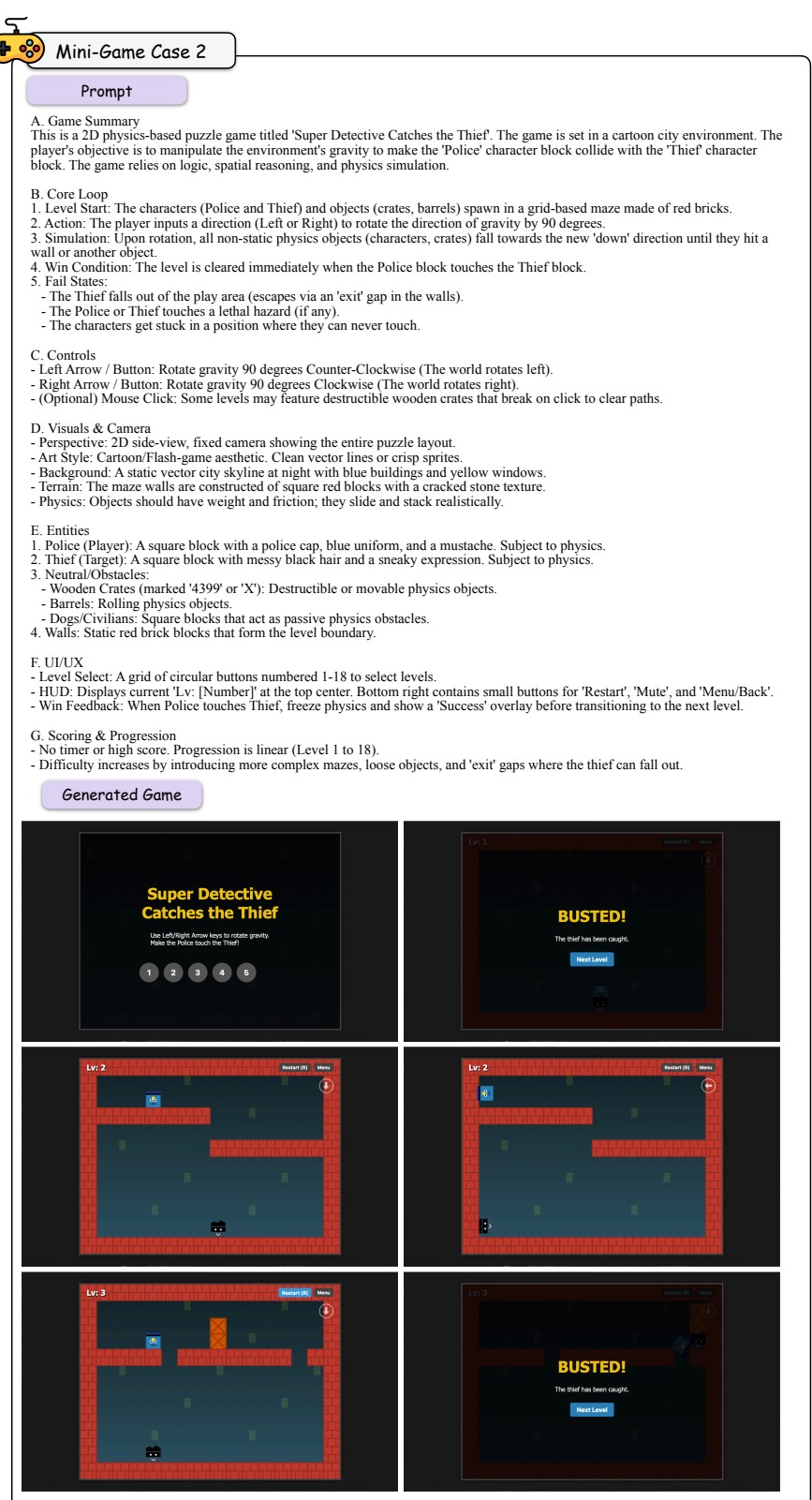

*Figure 10.* **Case 2: "Super Detective Catches the Thief" (Gravity-rotation physics puzzle).** ALIVE-Coder produces a 2D side-view puzzle layout built from static red brick walls over a night-city vector background. The world supports discrete gravity rotations via *Left/Right* controls (keyboard and on-screen buttons), after which all dynamic bodies (Police, Thief, crates, barrels) fall and settle under the new "down" direction with collision and stacking. The game terminates immediately upon Police–Thief contact with a *Success* overlay, and includes a level-select menu (1–18) and utility controls (*Restart*, *Mute*, *Menu/Back*) consistent with the prompt.

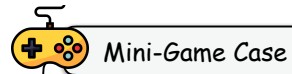

Mini-Game Case 3

**Prompt**

Create a 2D physics-based puzzle game titled 'Piggy Wants to Go Home'. The objective is to guide a piglet, who is suspended in the air by a red balloon, to a wooden exit door. The player does not control the pig directly but manipulates the environment (wind fans) to navigate obstacles.

Core Loop:
1. Level Start: The pig spawns at a starting point, floating upwards due to the balloon's buoyancy.
2. Navigation: The player clicks/holds specific fans placed in the level to generate wind. This wind pushes the pig in specific directions.
3. Objectives: The player must navigate the pig to collect up to 3 'food sack' items (encased in bubbles) and a Key (if present) to unlock barriers.
4. Fail State: If the balloon or pig touches hazards (spikes, sawblades), the balloon pops, and the level restarts.
5. Win State: The pig touches the wooden door area safely.

- Left Mouse Button: Click and hold on a fan to activate it. The fan spins and blows wind in its facing direction while held. Release to stop.
- R Key: Restart Level.
- P Key: Pause Game.

- Perspective: 2D side-scrolling view (fixed camera per level or scrolling if level is large).
- Art Style: Cartoon/Flash-game aesthetic. Bright colors, thick outlines.
- Background: Dark grey or greenish stone brick wall texture (dungeon style).
- Particle Effects: Wind streams when fans are active; 'Pop' effect for balloon; sparkles when collecting items.
- Player (Pig): A pink cartoon pig wearing blue overalls and a yellow belt. Attached by a yellow string to a bright red round balloon. The entity has physics properties: the balloon pulls up (buoyancy), the pig pulls down (gravity), creating a pendulum-like movement.
- Fans: White 3-blade propellers with red centers. Fixed positions on walls/air. Some levels have larger green fans.
- Hazards:
  - Spikes: Golden triangular spikes on floors, walls, or moving platforms. Lethal on contact.
  - Sawblades: Large, rotating grey circular saws. Lethal.
- Collectibles:
  - Food Sacks: Yellow sacks with red dots inside a transparent bubble.
  - Key: Golden key used to unlock 'Ice/Glass' barriers.
- Exit: A wooden frame door with a dark opening.:
- HUD: 'LEVEL [Number]' displayed at the bottom center. 'Time: [Seconds]' displayed at the top right.
- Menus: Simple Main Menu with 'PLAY' button and Level Select grid.
- Levels increase in difficulty with more spikes, moving platforms, and locked doors.
- Score is determined by the number of sacks collected (0-3) and time taken.

**Generated Game**

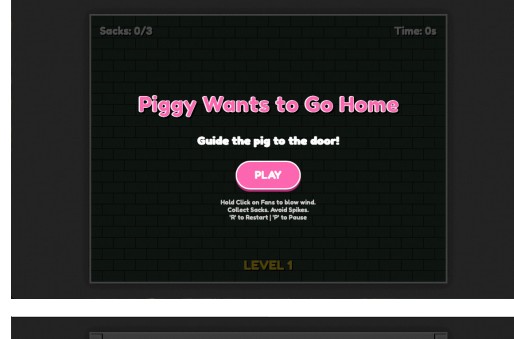
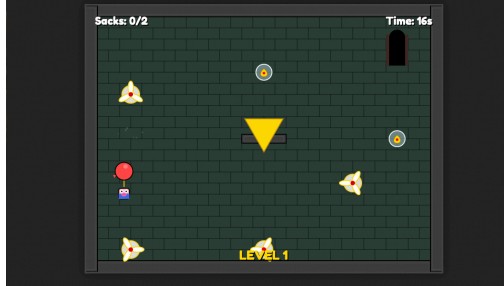
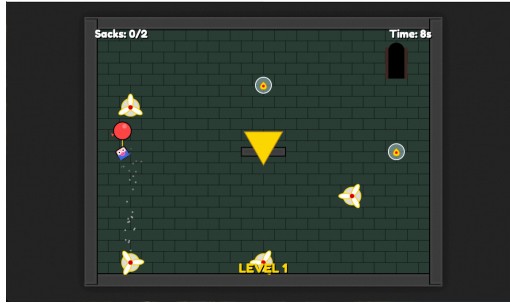
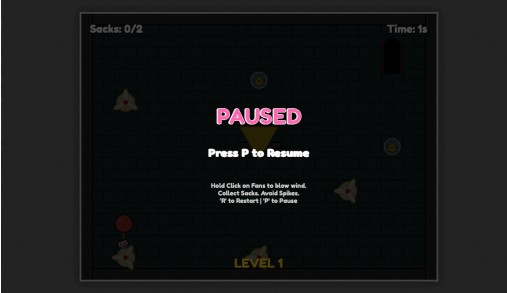

*Figure 11.* **Case 3: "Piggy Wants to Go Home" (Balloon buoyancy with fan-controlled wind).** ALIVE-Coder generates a 2D physics environment in a dungeon-style brick backdrop, where a piglet tethered to a red balloon exhibits buoyancy–gravity coupling and pendulum-like motion. Navigation is realized through click-and-hold activation of directional fans that emit visible wind streams and impart force to the pig/balloon system. The implementation includes hazards (spikes and rotating sawblades) that pop the balloon on contact (restart), collectibles (bubble-encased food sacks and an optional key) that update progress/score, and a wooden exit door that triggers level completion. The HUD renders LEVEL [#] at the bottom center and Time:  [s] at the top right, with keyboard shortcuts for restart (R) and pause (P).

# L. Baseline: GUI-Agent Judge (Playwright ReACT Agent)

This appendix details the *GUI-agent judge* used as a baseline in our experiments. The baseline follows a multi-turn **ReACT** loop (Reasoning–Action–Observation) with an explicit **memory** field, and interacts with the rendered mini-game through a real browser controlled by **Playwright**. Unlike ALIVE (one-shot planning), this agent performs *serial* perception and decision making, incurring latency and token growth with trajectory length.

## L.1. Overview

Given a task prompt (the game specification) and the current page state, the agent repeatedly: (i) observes the environment (screenshots and lightweight textual signals), (ii) queries an LLM for the next action in a constrained action space, (iii) executes the action in Playwright, and (iv) logs the resulting observation and updates memory.

The baseline is **verification-oriented** only implicitly: it attempts to complete or probe the game through interaction, but does not synthesize an explicit checklist or one-shot plan as in ALIVE.

**State and logs.** For each run, the agent creates a session directory and stores: (a) per-step screenshots, (b) conversation snippets, (c) action traces, and (d) a running "reflection memory" (a persistent text buffer). To control context growth, it maintains a bounded *context window* and a *max history token* budget; older messages (especially image-bearing ones) are pruned.

## L.2. Action Space and Protocol

The agent uses a small, browser-grounded action space that can be executed deterministically by Playwright. Each LLM decision must output a JSON object with four fields: `thought`, `action`, `action_input`, and `memory`. The action set is:

- `click`: left click at current mouse position (optionally with modifiers or right-click).

- `move x,y`: move mouse to absolute viewport coordinates.

- `drag x,y`: drag from current position to target coordinates.

- `scroll_up amount` / `scroll_down amount`: scroll the page.

- `write text`: type into the focused element.

- `press_key key`: press a key (or a chord, e.g., `Control+KeyR`).

- `hold_key key[,duration]`: hold a key for a short duration.

**Coordinate system (Playwright).** All pointer actions use viewport pixel coordinates $(x, y)$. In our Playwright setup, the browser is launched with a fixed viewport (e.g., $1280 \times 720$) and device scale factor 1.0, and coordinates are interpreted in the `page.mouse` coordinate space. To reduce ambiguity, we optionally overlay a light grid on the page (CSS absolute overlay) during evaluation; however, unlike the reference implementation that assumes a DOS resolution and hard-coded ranges, our coordinates follow the Playwright viewport and are therefore game-agnostic.

## L.3. Agent Architecture

**Core components.** The baseline agent comprises:

1. **LLM Client**: performs multimodal inference given recent messages and screenshots.

2. **Playwright Executor**: maps abstract actions to Playwright API calls (`page.mouse`, `page.keyboard`, `page.evaluate`).

3. **History Manager**: stores (i) a bounded multimodal context for inference and (ii) a full text-only log for auditing.

4. **Reflection Memory**: a persistent text buffer updated by the model to retain long-horizon information.

**Message representation and pruning.** Each message contains `role`, `content`, and a flag `has_image`. The agent keeps only the last $K$ messages (`context_window`) and additionally prunes until an approximate token budget is satisfied. Image messages are pruned more aggressively to prevent a high ratio of screenshots dominating the context.

### L.4. Per-Step Interaction Loop

Algorithm L.4 describes the baseline loop used to interact with the rendered game.

**GUI-Agent Judge (Playwright ReACT Loop)** **Input:** Task prompt $P$, maximum steps $T$, Playwright page $\mathcal{E}$, LLM $\mathcal{M}$.

1. Initialize history $\mathcal{H} \leftarrow [\text{SYSTEM}(P)]$, reflection memory $m \leftarrow \emptyset$.

2. For $t = 1$ to $T$:

   (a) Capture screenshot(s) $S_t \leftarrow \text{RENDER}(\mathcal{E})$.

   (b) Append observation to history:
   $$\mathcal{H} \leftarrow \mathcal{H} \cup [\text{USER}(\text{"Frame"}, S_t)].$$

   (c) Prune $\mathcal{H}$ to satisfy context window and token budget.

   (d) Construct query with reflection prompt and memory:
   $$Q_t \leftarrow [\mathcal{H}, \text{USER}(\text{REFLECT}(m))].$$

   (e) Query the LLM: $R_t \leftarrow \mathcal{M}(Q_t)$.
   *Note: $R_t$ is a JSON object with fields: thought, action, action_input, memory.*

   (f) Parse $(a_t, u_t, m_t) \leftarrow \text{PARSE}(R_t)$.

   (g) If $m_t \neq \emptyset$, set $m \leftarrow m_t$.

   (h) Append assistant message:
   $$\mathcal{H} \leftarrow \mathcal{H} \cup [\text{ASSISTANT}(R_t)].$$

   (i) Execute action:
   $$\text{PLAYWRIGHTEXEC}(\mathcal{E}, a_t, u_t).$$

   (j) Collect lightweight observation info (optional): console errors, DOM focus, URL, etc.

**Observation packaging.** At each step, the agent encodes the screenshot(s) as base64 data URLs and inserts them into a multimodal user message. If multiple frames are collected (e.g., before/after action, or short bursts), they are appended as separate image entries. This mirrors a "trajectory snippet" but remains within a bounded window to control cost.

**Reflection memory update.** The LLM may update the `memory` field to persist critical information (e.g., discovered controls, win conditions, or failure hypotheses). Memory overwrites are permitted; therefore, the prompt explicitly instructs the model to carry forward prior memory if it should be retained.

### L.5. Playwright Execution Details

**Browser instantiation.** We launch Chromium via Playwright with deterministic settings: fixed viewport, disabled animations where possible, and a per-task isolated browser context to avoid state leakage (cookies, local storage). Each generated game is loaded as a local HTML file or served from a local HTTP endpoint. We enable console and page error listeners to record runtime failures, which are stored as part of the trace for auditing.

**Action implementation.**

- **Pointer actions** use `page.mouse.move`, `page.mouse.down/up`, and `page.mouse.click`.

- **Keyboard actions** use `page.keyboard.press` for chords and `page.keyboard.down/up` for holds.

- **Scrolling** uses `page.mouse.wheel` with a signed delta proportional to `amount`.

- **Typing** uses `page.keyboard.type` with a modest delay to improve compatibility with in-game input handlers.

**Timing.** After each action, the agent waits for a short fixed delay and optionally for `requestAnimationFrame` ticks to allow the game to update. This is crucial for mini-games implemented with Canvas/WebGL where DOM events do not immediately reflect state changes.

## L.6. Discussion: Why This Baseline is Expensive

This baseline is representative of "agent-as-a-judge" evaluation: it is flexible and can uncover interactive failures, but it requires: (i) multimodal inference at each step, (ii) accumulating context (screenshots and text), and (iii) long-horizon exploration to expose mechanics. Consequently, its cost grows approximately linearly with interaction steps, and reliability is sensitive to compounding localization errors (mis-clicks) and action hallucinations. These limitations motivate ALIVE's decoupled one-shot planning and execution-based verification.

