# OpenReview forum: "Bringing Code ALIVE: Optimizing Interactive Frontend Mini-Games via Automated Play and Reinforcement Learning at Scale"
_ICML.cc/2026/Conference — ICML 2026 regular_

### Official Review · Reviewer_5pp4 · 2026-02-22

**Soundness:** 3
**Presentation:** 3
**Significance:** 4
**Originality:** 3
**Overall Recommendation:** 5
**Confidence:** 3

**Summary:**

This paper introduces ALIVE (Aligning LLMs via Interactive Visual Execution), a scalable evaluation-and-training framework for interactive frontend mini-game generation. ALIVE generates a task-specific + universal checklist and uses an MLLM to produce a one-shot action plan from the prompt, code, DOM, and accessibility tree; a lightweight browser executor (Playwright) runs the actions to collect an interaction trace, which a judge scores to produce an ALIVE-Score, with automatic replanning when plans are infeasible. Experiments show ALIVE aligns substantially better with human judgments than static code or single-frame visual judges while approaching GUI-agent accuracy at far lower cost, and it enables building the ALIVE-MiniGame dataset and training ALIVE-Coder via SFT and GRPO using ALIVE-Score as an execution-grounded reward, yielding strong gains on ALIVE-Eval and related benchmarks.

**Compliance With Llm Reviewing Policy:**

Affirmed.

**Final Justification:**

The paper is thorough and content-rich, with substantial technical details and strong empirical support, and I am comfortable recommending acceptance.

**Key Questions For Authors:**

1. Have you evaluated ALIVE (and ALIVE-Coder) beyond mini-games on more general web apps (multi-page flows, forms, async/network-heavy UIs, drag-and-drop, auth), and if not, can you provide a small pilot study or representative failures?
2. Can you quantify where ALIVE mistakes come from (planner grounding vs execution noise vs judge errors vs checklist gaps), especially on real-time/continuous-control games where one-shot planning may be brittle?
3. Since ALIVE-Score is both the main metric and the RL reward, how robust are the gains under “out-of-evaluator” changes (different judge backbones/prompts, browser/environment settings, or a different checklist generator/task family)?

**Limitations:**

yes

**Strengths And Weaknesses:**

**Strengths**
1. ALIVE focuses on what really matters for frontend “vibe coding”—runtime interactivity—which static code or single-frame visual judges often miss, while staying scalable via one-shot planning + lightweight execution for large-scale evaluation/data curation.
2. The planner makes good use of code + DOM + accessibility signals to craft targeted interaction sequences, and the feasibility checks + replanning help avoid false negatives from bad action grounding.
3. The empirical section is convincing and, importantly, the framework proves useful beyond evaluation by enabling SFT filtering and serving as an RL reward (GRPO) that improves the model.

**Weakness**
1. Most results are on frontend mini-games; it’s still unclear how well ALIVE generalizes to broader web apps (multi-page flows, async/network-heavy UIs, drag-and-drop, auth, etc.).
2. The paper could do a better job breaking down failure sources (planner vs executor vs judge vs checklist), especially for real-time/continuous-control games where one-shot plans may be brittle.
3.Several reproducibility-critical details are thin in the main paper (checklist/action-list formats, trace contents, invalid-plan rules, replanning constraints, canvas-click inference).
4. Because ALIVE-Score is both the key metric and the RL reward produced by LLM, there’s some risk of “teaching to the evaluator”; stronger out-of-evaluator checks (different judges/environments/task families) would strengthen the claims.

---

> ### Author Rebuttal · Authors · 2026-03-31
>
> We sincerely thank Reviewer 5pp4 for recognizing the value of our work and providing constructive feedback. Below are our detailed responses to your questions and concerns.
> ### [W1/Q1] Generalization to Complex Web Applications and Focus on Mini-Games
> We intentionally focus on mini-games as a controlled testbed because they cleanly expose the shortcomings of static analysis (e.g., missing event bindings) during runtime interaction. Unlike existing static web datasets (e.g., image-to-HTML), mini-games demand high interactivity while avoiding the confounding factors of complex web apps (e.g., authentication, persistent states).
>
> To address your concern regarding generalization, we conducted a new pilot study on 10 prominent product-level websites (e.g., apple.com), manually mocking their JSON backends. We utilized ALIVE to generate checklists and evaluate interactions, achieving high consistency with human judgment. However, this study highlighted two primary challenges for evaluating complex web apps:
>
> 1. Longer Action Horizons: The action lists are significantly longer, requiring extensive navigation and button clicks, which necessitates segmenting inputs for VLMs.
> 2. Backend Dependency: Complex apps cannot be fully decoupled from mocked backends, shifting the focus away from pure frontend interactive evaluation.
>
> Our primary goal in this paper is to establish a scalable evaluation framework in a self-contained, high-interaction setting. Extending ALIVE to complex web applications is an important direction for future work. We will include this pilot study and its insights in the revised manuscript.
> ### [W2/Q2] Explicit Failure Analysis and Reproducibility
> Reproducibility: All pipeline details are fully transparent in our appendices (**Appendix B**: checklist JSONs, **Appendix C**: trace contents/invalid-plan rules, **Appendix D**: replanning constraints, **Appendix J**: system prompts). We will explicitly refine and signpost these sections in the revision to ensure clarity on canvas-click inference and plan validation.
>
> Failure Analysis: We manually analyzed a representative subset of 200 failure cases from the **Section 3** dataset (due to the time limitation). We categorize ALIVE’s failures into five main types:
> 1. Planner Limitations (~33%): Runtime-adaptive tasks depending on intermediate states.
> 2. Planner Grounding (~21%): Weak DOM/accessibility grounding in canvas-dominant games.
> 3. Checklist Gaps (~18%): Imperfect checklist generation or specification gaps.
> 4. Execution Noise/Limits (~16%): Continuous-control and timing-sensitive interactions.
> 5. Judge Errors (~12%): Delayed-effect or long-horizon verification issues.
>
> We will add this taxonomy and preliminary findings to the revision, and commit to a full-scale study in the camera-ready version. Importantly, our replanning mechanism already resolves many initial grounding errors. Failure rates dropped substantially for challenging genres like Strategy (30.8% $\rightarrow$ 10.1%), Puzzle (24.5% $\rightarrow$ 7.5%), and Parkour (23.8% $\rightarrow$ 7.2%). This proves that many errors stem from imperfect first-pass grounding rather than evaluation inability, narrowing residual failures to tasks strictly requiring closed-loop agents.
> ### [W3/Q3] "Teaching to the Evaluator" / Circular Reasoning
> We agree that if a reward solely improves performance on the same judge without translating to human preference, it would weaken the validity of the evaluation. However, the improvements demonstrated by ALIVE-Coder do not exploit judge biases, owing to the following reasons:
>
> 1. Execution-Grounded Evaluation: The ALIVE reward is not a pure LLM language preference score. It strictly requires browser execution, interaction trace collection, DOM-grounded action plans, and post-hoc checklist scoring. To achieve a high score, the model cannot rely on superficial text patterns; it must generate code where buttons are genuinely clickable and target states are successfully reached.
> 2. Proven Generalization and Objectivity: As detailed in **Section 3** and **Figure 4**, the ALIVE-Judge exhibits significantly higher alignment with human judgment across 4 different backbone LLMs compared to static code and vision judges, confirming its objectivity and demonstrating sufficient robustness across different models.
>
> 3. Out-of-Evaluator Validation: While we did not re-run the full RL pipeline with different judge backbones due to high computational constraints, our human evaluation (**Section 4.4.2**) and the independent **ArtifactsBench** results serve as our primary out-of-evaluator validation. The results show a substantial increase in human preference following SFT and RL, and the notable performance gains on Artifacts corroborate the broader effectiveness of our pipeline beyond our specific judge or checklist generator.

---

> > ### Author Rebuttal · Reviewer_5pp4 · 2026-04-01
> >
> > Thank you for the detailed rebuttal and additional experiments. My main concerns have been sufficiently addressed, and I will raise my score.

---

> > > ### Author Response · Authors · 2026-04-04
> > >
> > > Dear Reviewer 5pp4,
> > >
> > > Thank you so much for updating your score and for your continued support! We are thrilled that our new pilot study on complex web applications successfully addressed your concerns.
> > >
> > > We deeply appreciate your recognition of ALIVE's core focus on runtime interactivity and its value for the community. Thank you again for your time and the highly constructive insights that helped us strengthen this work.

---

### Official Review · Reviewer_c8Hu · 2026-03-12

**Soundness:** 3
**Presentation:** 3
**Significance:** 3
**Originality:** 3
**Overall Recommendation:** 5
**Confidence:** 3

**Summary:**

This paper introduces ALIVE, a framework for efficiently evaluating  LLM-generated interactive front-end games. The framework follows a plan-execute-evaluate paradigm, where the model first produce a list of actions for testing the game given a hybrid criteria and game structure analyzed in a preprocessing phase, the execution phase uses a browser engine to perform the planned actions and saves the trajectory, which is provided to the judge to assign a score. The evaluation results on the generation rank and score are aligned with human evaluation. The efficiency is improved compared to interactive computer-use agents. The evaluation framework is further utilized to select SFT data and provide RL rewards.

**Compliance With Llm Reviewing Policy:**

Affirmed.

**Final Justification:**

The rebuttal addressed my main concerns therefore I raised the score

**Key Questions For Authors:**

See weaknesses, besides,
1. What is the rank agreement and score precision of GUI-agent judge compared to human baseline?

**Limitations:**

yes

**Strengths And Weaknesses:**

Strengths:
- Decompose the reasoning from execution to improve efficiency, exploring the middle ground between static evaluation and agent-based interaction.
- The evaluation framework also help curate training data with better quality and provide reward signal for non-verifiable problems

Weaknesses:
- The motivation of evaluating only mini-game generation is not clear, why not include other general interactive applications? It's also unclear that if the one-shot planning method can generalize to other visual application development tasks.
- The plan-execute paradigm is unable to adapt to more dynamic executions, some problems / edge cases might be only discoverable during the game runtime, the pre-planned execution trace will miss critical issues in those intermediate game states.
- ALIVE-Coder-RL is trained using ALIVE as a reward signal and is evaluated on the same benchmark, which makes the performance gain less impressive.

---

> ### Author Rebuttal · Authors · 2026-03-31
>
> We sincerely thank the reviewer for recognizing our framework's efficiency, as well as its contribution to curating high-quality training data and providing reliable reward signals. Below, we address your comments in detail.
>
> ### [W1] Motivation for focusing on mini-games rather than general web applications.
> We focus on interactive mini-games because they serve as a controlled, highly representative testbed for evaluating runtime interactivity—a pressing challenge in the community where a significant capability gap remains between open-source and proprietary models. Existing public datasets primarily target simpler, static web generation (e.g., image-to-HTML) where complex interaction is rarely required and static judges already perform adequately.
>
> Conversely, mini-games cleanly expose the critical shortcomings of static analysis (e.g., missing event bindings, broken logic) that only surface during runtime. They retain rich, heavy event-driven behavior while avoiding the confounding factors of broader web apps, such as authentication, persistent user states, and complex backend mocking. Extending ALIVE to complex, product-level web applications and developing corresponding highly-interactive public datasets are important directions for our future work. Due to space constraints, please refer to our response to Reviewer 5pp4 for details on a new pilot study extending ALIVE to complex, product-level web applications.
>
> ### [W2] Limitations of the plan-execute paradigm in dynamic intermediate states.
> We agree that a more explicit failure analysis strengthens the paper. To clarify this, we categorize ALIVE’s failures into five main types:
>
> 1. Runtime-adaptive tasks where the next action depends on intermediate states.
> 2. Continuous-control and timing-sensitive interactions.
> 3. Weak DOM/accessibility grounding in canvas-dominant games.
> 4. Delayed-effect or long-horizon verification issues.
> 5. Checklist/specification gaps.
>
> We will add this taxonomy and representative examples to the revision. Importantly, our replanning mechanism largely mitigates these issues. As detailed in **Appendix D**, a clear pattern emerges: lightweight replanning recovers a substantial portion of initial failures for challenging, high-error genres. For example, failure rates dropped significantly for Strategy (30.8% to 10.1%), Puzzle (24.5% to 7.5%), and Parkour (23.8% to 7.2%). This indicates that many errors stem from imperfect first-pass grounding rather than an inability to evaluate the task, narrowing the remaining failures strictly to settings that require closed-loop agents.
>
> ### [W3] Concerns regarding training and evaluation circularity.
> The improvements demonstrated by ALIVE-Coder are not the result of "circular reasoning" or exploiting judge biases, for three key reasons:
>
> 1. Execution-Grounded: The ALIVE reward is not a pure LLM language preference score. It strictly requires browser execution, interaction trace collection, and DOM-grounded action plans. To achieve a high score, the model must generate code where buttons are genuinely clickable, events trigger correctly, and target states are actually reached.
>
> 2. Proven Generalization: As shown in **Section 3**, the ALIVE-Judge exhibits significantly higher alignment with human judgment across multiple backbone models compared to static judges, confirming its objectivity.
>
> 3. Out-of-Evaluator Validation: To ensure our training improvements extend beyond the ALIVE-Eval metric, we conducted extensive human evaluations (**Section 4.4.2, Figure 7**), showing a substantial increase in human preference following SFT and RL. Furthermore, we observe notable performance gains on the independent **Artifactsbench**.
>
> ### [Q1] Rank agreement and score precision of the GUI-agent judge.
> Subject to the prohibitive token consumption of running iterative GUI agents on expensive backbone LLMs, we were unable to conduct a complete GUI judge experiment across all models. However, we evaluated a GUI-agent baseline using Gemini-3-Flash with a 20-agent-step limit.
>
> On the same **Section 3** test set, the GUI judge achieves a rank agreement of 0.77 and a score precision of 0.92 compared to the human baseline. As detailed in our cost analysis (**Appendix F**), ALIVE is highly competitive and efficient:
>
> * ALIVE (max replan = 1) matches a GUI-agent judge operating at 9–10 steps.
>
> * ALIVE (max replan = 3) matches a GUI-agent judge operating at 11–12 steps.

---

> > ### Author Rebuttal · Reviewer_c8Hu · 2026-04-03
> >
> > I thank the authors for their detailed response. My concerns have been fully addressed and I will accordingly increase the score.

---

> > > ### Author Response · Authors · 2026-04-04
> > >
> > > Dear Reviewer c8Hu,
> > >
> > > Thank you so much for your time and the positive confirmation! We are very glad that our responses have fully resolved your concerns.
> > >
> > > We noticed that your official score in the OpenReview system hasn't reflected the update yet. Just a gentle reminder in case it was overlooked. We truly appreciate your support and your constructive feedback!

---

### Official Review · Reviewer_kS4w · 2026-03-13

**Soundness:** 3
**Presentation:** 3
**Significance:** 3
**Originality:** 3
**Overall Recommendation:** 4
**Confidence:** 3

**Summary:**

The paper proposes ALIVE, a high-throughput evaluation framework designed to assess the quality of LLM-generated interactive frontend mini-games. The authors identify a critical bottleneck in evaluating frontend code: static analysis and single-frame visual inspection fail to verify dynamic interactivity, while fully autonomous GUI agents are computationally prohibitive. To address this, ALIVE utilizes a one-shot planning mechanism that analyzes the Document Object Model (DOM) and Accessibility Tree to synthesize a sequence of interactions. These interactions are then executed via a headless browser, and the resulting visual trace is evaluated against functional and aesthetic checklists. The authors demonstrate that ALIVE achieves high correlation with human judgment at a fraction of the computational cost of iterative GUI agents, and prove its usefulness as a reward signal.

**Compliance With Llm Reviewing Policy:**

Affirmed.

**Key Questions For Authors:**

- The framework relies on one-shot planning, which seems suitable for relatively predictable mini-games. How would this approach scale to complex web applications (e.g., SaaS dashboards, multi-step forms) where the required actions depend heavily on intermediate dynamic states or backend responses that cannot be anticipated in a single planning pass?
- The ALIVE-Coder model shows significant gains on the ALIVE-Eval benchmark. Given that the model was optimized using the ALIVE framework as a reward signal, how do you address concerns that the model is simply overfitting to the specific biases and evaluation patterns of the ALIVE judge?

**Limitations:**

yes

**Strengths And Weaknesses:**

# Soundness

Strengths
- The experimental methodology is generally robust and well-designed
- The efficiency analysis provides a compelling cost-benefit justification for the one-shot planning approach, detailing significant reductions in token consumption and latency compared to iterative GUI agents
- The analysis of RL training dynamics provides valuable insights

Weaknesses
- The ALIVE-Eval score is generated using the exact same evaluation pipeline that provided the reward signal during RL training. While human evaluation corroborates the results, relying heavily on a self-proposed metric for the main claim is questionable
- The dataset construction relies exclusively on Gemini-3-Pro for generating SFT candidates, which may introduce a bias or limit the applicability of the model

# Presentation

Strengths:
- The paper is well-structured, and the narrative flows logically from problem identification to the proposed solution and empirical validation
- The motivation is clearly illustrated through compelling examples (e.g., Figure 1 and Figure 2), which effectively highlight the limitations of existing evaluation paradigms

# Significance

Strengths:
- The paper addresses a highly relevant and growing problem in the field of code generation: the reliable and scalable evaluation of interactive, visually rich applications
- The demonstration that execution-based feedback can lead to good results with RL is a significant contribution that could influence future research in aligning code-generation models

Weaknesses:
- The scope of the framework is currently limited to frontend HTML mini-games. While games are a good proxy for interactivity, it remains unclear how well the one-shot planning mechanism generalizes to more complex, stateful web applications (e.g., e-commerce platforms or data dashboards) where interaction trajectories are highly dependent on backend responses and complex user states

# Originality

Strengths
- The combination of one-shot planning, DOM/Accessibility Tree analysis, and automated execution for evaluating generated code represents a novel synthesis of existing techniques
- The application of this interactive verification pipeline as a scalable reward signal for RL in the context of frontend generation is a relatively new perspective that goes beyond static unit testing

Weaknesses
- The individual components of the framework are not novel in themselves
- The approach is conceptually similar to recent "Agent-as-a-Judge" paradigms, although the one-shot planning adaptation provides a distinct efficiency advantage

---

> ### Author Rebuttal · Authors · 2026-03-31
>
> We sincerely thank you for recognizing the robustness of our experimental methodology, our efficiency analysis, and the novelty of combining one-shot planning with execution-based feedback for RL. Below, we address your specific concerns and questions.
>
> ### [W1, Q2] Overfitting to the ALIVE Judge and Circular Evaluation
> We agree that if a reward solely improves performance on the same judge without translating to human preference, it would weaken the validity of the evaluation. However, the improvements demonstrated by ALIVE-Coder are not the result of "circular reasoning" or exploiting judge biases, for the following reasons:
>
> 1. Execution-Grounded, Not Language-Biased: The ALIVE reward is not a pure LLM language preference score. It strictly requires browser execution, interaction trace collection, DOM/accessibility-grounded action plans, and post-hoc checklist-based scoring. To achieve a high score, the model cannot rely on superficial text patterns; it must generate code where buttons are genuinely clickable, events trigger correctly, and target states are successfully reached.
>
> 2. Proven Generalization and Objectivity: As demonstrated in **Section 3**, the ALIVE-Judge exhibits significantly higher alignment with human judgment across multiple backbone models compared to static code and vision judges, confirming its objectivity.
>
> 3. Out-of-Evaluator Validation: To ensure our training improvements extend beyond the ALIVE-Eval metric, we conducted extensive human evaluations (**Section 4.4.2, Figure 7**). The results show a substantial increase in human preference following SFT and RL. Furthermore, the notable performance gains on the independent **Artifactsbench** corroborate the broader effectiveness of our training pipeline.
>
> ### [Q1] Generalization to Complex Web Applications
> We intentionally focus on interactive mini-games because they provide a controlled yet highly representative testbed for evaluating runtime interactivity. Currently, existing public datasets primarily focus on simpler, static web generation tasks (e.g., image-to-HTML) where complex interaction is rarely required, static judges perform adequately, and the capability gap between models is narrow.
>
> Conversely, mini-games cleanly expose the critical shortcomings of static analysis (e.g., missing event bindings, broken logic) that only surface during runtime interaction. They retain rich event-driven behavior while avoiding the confounding factors of broader web apps, such as authentication, persistent user states, and complex backend mocking. Extending the ALIVE framework to handle the dynamic states of complex, product-level web applications (like SaaS dashboards) and developing corresponding highly-interactive public datasets are important directions for our future work. Our primary goal in this paper is to establish a scalable evaluation framework in a self-contained, high-interaction setting where the limitations of existing static judges are most pronounced. Due to space constraints, please refer to our response to Reviewer 5pp4 for details on a new pilot study extending ALIVE to complex, product-level web applications.
>
> ### [W2] Potential Bias from Gemini-3-Pro Data
> We mitigate the risk of teacher bias and model lock-in through three key design choices:
> 1. Execution-Based Filtering over Imitation: Gemini-3-Pro serves only as a candidate source, not as absolute ground truth. To reduce source bias, we generate five distinct candidate solutions for each query (**Section 4.2**). We then rely on execution-based filtering to select only functional, interactive samples. Furthermore, our RL stage depends entirely on environment feedback, rather than teacher imitation.
>
> 2. Utility-Driven Objective: The training objective is to maximize execution-grounded utility, not to replicate Gemini's coding style. If the model learns to mimic Gemini's stylistic patterns but fails to produce code that works in the browser, it will receive a low reward.
>
> 3. Strong Out-of-Distribution Generalization: The fact that ALIVE-Coder exhibits significant improvements across different prompt distributions, held-out tasks, and independent human evaluations demonstrates that it has learned robust interactive coding capabilities, rather than simply distilling the surface-level style of the candidate source.

---

### Official Review · Reviewer_pN4J · 2026-03-14

**Soundness:** 3
**Presentation:** 3
**Significance:** 3
**Originality:** 3
**Overall Recommendation:** 4
**Confidence:** 3

**Summary:**

The paper proposes ALIVE, a framework that decouples reasoning from execution for interactive frontend mini-game evaluation and optimization. Instead of relying on static code analysis or screenshots, ALIVE verifies generated games by planning actions, executing them in a browser, and scoring the observed interaction traces. This signal is further used to curate SFT data and provide rewards for RL. Experimental results suggest that ALIVE aligns more closely with human judgments than static judges while remaining substantially cheaper than GUI agents.

**Compliance With Llm Reviewing Policy:**

Affirmed.

**Key Questions For Authors:**

- While the paper focuses on mini-games, the underlying one-shot planning and DOM-based verification logic appears potentially generalizable to broader web interactions. I would be interested to see if the authors validated its effectiveness on complex, product-level frontend tasks.
- Could the authors consider validating the proposed framework on publicly available datasets?

**Limitations:**

yes

**Strengths And Weaknesses:**

Strengths
+ The paper is well motivated and studies an interesting problem. It identifies a meaningful gap in current frontend code generation research, namely that static judges are often insufficient for evaluating whether generated interactive artifacts are truly usable and functional.
+ The proposed framework is well structured, and the Plan–Execute–Evaluate pipeline is described in a clear and intuitive manner. The design choices are generally well explained, making the proposed approach accessible to readers.
+ The framework offers a reasonable trade-off between reliability and efficiency, which makes it appealing for realistic use in large-scale evaluation, data filtering, and reinforcement learning pipelines

Weaknesses
- The paper acknowledges that one-shot planning may struggle in highly real-time genres, but the current discussion remains brief. A more detailed failure analysis would strengthen the paper.
- The effectiveness of ALIVE appears to rely heavily on the completeness and correctness of the synthesized checklists. If the checklist fails to capture important aspects of functionality or visual quality, the resulting score may not faithfully reflect the true quality of the generated game. More analysis on checklist coverage and robustness would make the evaluation framework more convincing.
- The use of ALIVE for evaluation, data filtering, and reinforcement learning is meaningful, but the paper states that the raw source data cannot be publicly released due to copyright considerations. It probably limits reproducibility and makes it harder for others to independently verify the dataset construction process and the resulting training pipeline.

---

> ### Author Rebuttal · Authors · 2026-03-31
>
> We sincerely thank you for recognizing the motivation of our work and the efficiency of the ALIVE framework. We appreciate your constructive feedback and address your specific questions below.
>
> ### [W1] Failure Analysis of One-Shot Planning
> We agree that a more explicit failure analysis strengthens the paper. As detailed in **Appendix D**, we report per-genre replanning and residual rectification rates across 30 game categories. A clear pattern emerges: low-error genres (e.g., Card, Text Adventure) are mostly short-horizon and event-driven, whereas high-error genres (e.g., Strategy, Survival) require runtime adaptation or fine-grained timing.
>
> To clarify this, we categorize ALIVE’s failures into five main types:
>
> 1. Runtime-adaptive tasks where the next action depends on intermediate states.
>
> 2. Continuous-control and timing-sensitive interactions.
>
> 3. Weak DOM/accessibility grounding in canvas-dominant games.
>
> 4. Delayed-effect or long-horizon verification issues.
>
> 5. Checklist/specification gaps.
>
> We will add this taxonomy and representative examples to the revision to show that these failures are structured and consistent with the scope of one-shot planning.
>
> Furthermore, the gap between pre-plan failure and post-rectification failure highlights the effectiveness of our replanning mechanism. For challenging genres such as Strategy (30.8% $\rightarrow$ 10.1%), Puzzle (24.5% $\rightarrow$ 7.5%), and Parkour (23.8% $\rightarrow$ 7.2%), lightweight replanning recovers a substantial portion of initial failures. This indicates that many errors stem from imperfect first-pass grounding rather than an inability to evaluate the task, narrowing the remaining failures to settings that strictly require closed-loop agents.
>
> ### [W2] Dependence on Checklist Completeness
> We agree that checklist quality is critical. To mitigate brittleness, our design deliberately avoids a single template, instead combining universal criteria with task-specific criteria synthesized dynamically from prompt, code, and DOM signals.
>
> More importantly, ALIVE does not judge static code against the checklist; it applies the checklist to an execution trace obtained from actual browser interaction. This execution-grounding substantially reduces the false-positive failure mode of static judges that reward plausible-looking but non-functional code. While we fully acknowledge that no checklist can exhaustively cover all functional or aesthetic aspects, the strong human agreement in our experiments demonstrates that the resulting signal is significantly more faithful than static alternatives and highly stable for SFT filtering and RL. We will expand our discussion on checklist robustness and limitations in the revision.
>
> ### [W3 & Q2] Copyright Constraints, Reproducibility, and Public Datasets
> To ensure full reproducibility without copyright infringement risks, we are actively processing the dataset. We will release a sanitized, open-source version of the benchmark instructions (with sensitive IPs removed). Additionally, we will fully open-source the evaluation pipeline, prompts, and training configurations so the community can independently verify our data construction and RL pipeline.
>
> ### [Q1 & Q2] Focus on Mini-Games, Broader Web Applications, and Public Datasets
> We intentionally focus on interactive mini-games because they provide a controlled yet highly representative testbed for evaluating runtime interactivity. Regarding public datasets, there is currently a lack of benchmarks that require complex interactive logic. Existing public datasets primarily focus on simpler, static web generation tasks (e.g., image-to-HTML) where complex interaction is rarely required, static judges already perform adequately, and the capability gap between models is relatively narrow.
>
> Conversely, mini-games cleanly expose the critical shortcomings of static analysis (e.g., missing event bindings, broken game logic) that only surface during runtime interaction. They retain rich event-driven behavior while avoiding the confounding factors of broader web apps, such as authentication, persistent user states, and complex backend mocking. Extending ALIVE to complex, product-level web applications and developing corresponding highly-interactive public datasets are important directions for our future work. Our primary goal in this paper is to establish a scalable evaluation framework in a self-contained, high-interaction setting where the limitations of existing static judges are most pronounced. Due to space constraints, please refer to our response to Reviewer 5pp4 for details on a new pilot study extending ALIVE to complex, product-level web applications.

---

> > ### Author Rebuttal · Reviewer_pN4J · 2026-04-05
> >
> > Thanks for the rebuttal. I will maintain the score.

---

### Decision · Program_Chairs · 2026-04-30

**Decision:**

Accept (regular)

**Comment:**

This paper introduces ALIVE, a high-throughput plan-execute-evaluate framework for LLM-generated interactive frontend mini-games. It synthesizes a task-specific checklist from prompt, code, DOM and accessibility signals, generates a one-shot interaction plan, executes it in a headless browser, and scores the resulting trace. The score is used as a scalable pre-flight filter for SFT data curation and as an execution-grounded reward for RL, yielding ALIVE-Coder with strong gains over static-judge baselines.

All four reviewers lean positive after rebuttal (two Accept, two Weak Accept maintained).

Consensus strengths: well-motivated problem, clear identification that static analysis misses runtime interactivity, clean pipeline design with good efficiency trade-off versus GUI agents, thorough empirical validation showing higher human alignment, and demonstrated utility beyond evaluation for data filtering and RL.

Main weaknesses raised: limited scope to mini-games with unclear generalization to complex web apps, dependence on checklist completeness, risk of circular evaluation since ALIVE-Score is both metric and RL reward, brief initial failure analysis for real-time genres, reliance on proprietary model for SFT candidates, and limited reproducibility due to copyright constraints on raw data.